# HIF-1α Causes LCMT1/PP2A Deficiency and Mediates Tau Hyperphosphorylation and Cognitive Dysfunction during Chronic Hypoxia

**DOI:** 10.3390/ijms232416140

**Published:** 2022-12-17

**Authors:** Ling Lei, Jun Feng, Gang Wu, Zhen Wei, Jian-Zhi Wang, Bin Zhang, Rong Liu, Fei Liu, Xiaochuan Wang, Hong-Lian Li

**Affiliations:** 1Key Laboratory of Education Ministry/Hubei Province of China for Neurological Disorders, Tongji Medical College, School of Basic Medicine, Huazhong University of Science and Technology, Wuhan 430030, China; 2Co-Innovation Center of Neuroregeneration, Nantong University, Nantong 226001, China; 3Department of Neurosurgery, Union Hospital, Tongji Medical College, Huazhong University of Science and Technology, Wuhan 430022, China; 4Department of Neurochemistry, Inge Grundke-Iqbal Research Floor, New York State Institute for Basic Research in Developmental Disabilities, Staten Island, NY 10314, USA; 5Shenzhen Research Institute, Huazhong University of Science and Technology, Shenzhen 518000, China

**Keywords:** chronic hypoxia, HIF-1α, leucine carboxyl methyltransferase 1, protein phosphatase 2A, tau, cognitive impairments

## Abstract

Chronic hypoxia is a risk factor for Alzheimer’s disease (AD), and the neurofibrillary tangle (NFT) formed by hyperphosphorylated tau is one of the two major pathological changes in AD. However, the effect of chronic hypoxia on tau phosphorylation and its mechanism remains unclear. In this study, we investigated the role of HIF-1α (the functional subunit of hypoxia-inducible factor 1) in tau pathology. It was found that in Sprague-Dawley (SD) rats, global hypoxia (10% O_2_, 6 h per day) for one month induced cognitive impairments. Meanwhile it induced HIF-1α increase, tau hyperphosphorylation, and protein phosphatase 2A (PP2A) deficiency with leucine carboxyl methyltransferase 1(LCMT1, increasing PP2A activity) decrease in the rats’ hippocampus. The results were replicated by hypoxic treatment in primary hippocampal neurons and C6/tau cells (rat C6 glioma cells stably expressing human full-length tau441). Conversely, HIF-1α silencing impeded the changes induced by hypoxia, both in primary neurons and SD rats. The result of dual luciferase assay proved that HIF-1α acted as a transcription factor of LCMT1. Unexpectedly, HIF-1α decreased the protein level of LCMT1. Further study uncovered that both overexpression of HIF-1α and hypoxia treatment resulted in a sizable degradation of LCMT1 via the autophagy–-lysosomal pathway. Together, our data strongly indicated that chronic hypoxia upregulates HIF-1α, which obviously accelerated LCMT1 degradation, thus counteracting its transcriptional expression. The increase in HIF-1α decreases PP2A activity, finally resulting in tau hyperphosphorylation and cognitive dysfunction. Lowering HIF-1α in chronic hypoxia conditions may be useful in AD prevention.

## 1. Introduction

Alzheimer’s disease (AD) is histopathologically characterized by extracellular senile plaques consisting of β-amyloid (Aβ) and intracellular neurofibrillary tangles made up of the abnormally hyperphosphorylated tau [1,2,3,4]. In the clinic, AD presents as progressive cognitive impairments. In less than 5% of cases, the disease co-segregates with certain mutations in β-amyloid precursor protein, presenilin-1, or presenilin-2 genes [5]. More than 95% of AD patients belong to sporadic cases. Previous studies have proposed that due to multifactorial nature of this disease, different signaling pathways, through different disease mechanisms, apparently lead to the same two characteristic lesions of the disease, neurofibrillary degeneration due to abnormally hyperphosphorylated tau and β-amyloidosis [3,6,7]. 

Hypoxia is the most common pathological process resulting from inadequate oxygen supply to tissue or the inability to utilize oxygen by mitochondria. Accumulating evidence suggests that hypoxia is a risk factor and is closely associated with AD [8,9,10,11,12,13,14]. Senile plaques are predominantly composed of Aβ generated from amyloid precursor protein (APP) via the amyloidogenic pathway [15], by the sequential cleavage of APP by β-secretase (BACE1) and γ-secretase, with BACE1 as the rate-limiting enzyme in this process [16,17]. Previous studies showed that hypoxia facilitates Aβ production by up-regulating BACE1 gene expression [18,19,20] and promotes the accumulation of Aβ in exosomes by enhancing the interaction between CD147 and Hook1 [21]. Li et al. found that hypoxia increases the γ-secretase cleavage activity via upregulation of the expression of APH-1a, a component of the γ-secretase complex required for normal γ-secretase assembly [22]. Meanwhile, another study showed that hypoxia leads to an impairment of Aβ clearance via inhibition of the Aβ-degrading enzyme neprilysin (NEP) [23]. These researches obviously prove that hypoxia promotes Aβ generation and decreases its clearance, aggravating the pathological progression of AD. 

Hypoxia is also implicated in tau pathology, the other neuropathological alteration of AD. Chronic intermittent hypoxia was reported to enhance pathological tau seeding, propagation, and accumulation, and exacerbate Alzheimer-like memory and synaptic plasticity deficits and molecular signatures [14]. It is well known that tau pathology mainly results from tau hyperphosphorylation [24,25]. In the process of AD, the imbalance of tau kinases and phosphatases is the major cause of abnormal phosphorylation of tau [26,27]. However, the mechanism of the effect of hypoxia on tau hyperphosphorylation and cognitive impairment is still poorly understood.

Hypoxia-inducible factor 1 (HIF-1), a heterodimer composed of the constitutive subunit HIF-1β and the functional subunit HIF-1α, is an important transcription factor involved in hypoxia response [28]. In the presence of oxygen, HIF-1 is degraded by ubiquitin proteases. With hypoxia, the HIF-1α subunit cannot be degraded and accumulates, and binds to HIF-1β to form dimers, which are then transferred to the nucleus and activate genes encoding proteins involved in hypoxic homeostatic response [29].

To address the effect and mechanism of hypoxia in AD tau pathology, in the current study, we focused on the changes of HIF-1α and the downstream events with hypoxia treatment. We found that in Sprague-Dawley (SD) rats, chronic hypoxia induces cognitive impairment, and an increase in HIF-1α, tau hyperphosphorylation, and Protein Phosphatase 2A (PP2A) deficiency with a decrease of leucine carboxyl methyltransferase 1 (LCMT1, responsible for PP2A methylation and activity) in the hippocampus. Conversely, downregulation of HIF-1α attenuates hypoxia-induced tau pathologies and cognitive impairment. Hence, our findings imply that HIF-1α downregulating PP2A activity and promoting tau phosphorylation might be a mechanism-based therapeutic target for treating chronic hypoxia-related cognitive deficits including in AD.

## 2. Results

### 2.1. Chronic Hypoxia Impairs Cognitive Functions in Rats

Hypoxia is believed to be a risk factor for AD, and cognitive dysfunction is the distinct clinical presentation of AD. To investigate the effect of chronic hypoxia on cognitive function, twenty-four healthy SD rats were randomly divided into a hypoxia group treated with 10% O**_2_** for 6 h per day and a control group with normoxia (21% O**_2_**). After 30 days, we performed a series of behavioral tests (Figure 1). Figure 1a showed the 30 days course of the experiment from the start of treatment to the behavioral tests. Firstly, the open field test (OFT) results showed that there was no significant difference in the total distance covered between the two groups, supporting that hypoxia rats displayed similar patterns of anxiety and exploratory activity as control animals (Figure 1b). The novel object recognition (NOR) experiment helps to examine the short-term memory capacity. Our results from NOR experiment revealed that in the hypoxia group, the time for curiosity toward exploring new things was significantly reduced compared to that in control rats (Figure 1c). Next, we tested memory and learning ability using a Morris water maze (MWM) test, and observed that, compared with the control group, the hypoxia rats showed significantly increased latency to find the hidden platform (Figure 1d). On day 6, the spatial memory was tested by removing the platform. We observed a remarkable decrease in the latency to find the platform position as well as the crossing times of the platform position in the target quadrant in the hypoxia rats as compared to the control group (Figure 1e–g), while there was no difference in the swimming speed between the two groups (Figure 1h). Together, these findings indicate that chronic hypoxia causes learning and memory impairments in rats.

### 2.2. Chronic Hypoxia Leads to Tau Hyperphosphorylation Accompanied by LCMT1-Related Decrease in PP2A Activity in Rats’ Hippocampus

One of the hallmarks of Alzheimer’s disease is tau hyperphosphorylation, which is associated with neuronal insult and cognitive impairment [30,31]. The phosphorylation status of tau is tightly controlled by a delicate balance between protein kinases’ and phosphatases’ activities. Among the many brain protein kinases that regulate tau, by far, glycogen synthase kinase-3β (GSK-3β) is the major one [32]. Of particular relevance, the phosphatase battlefront is largely led by a distinct pool of protein phosphatase 2A (PP2A) enzymes that are responsible for the bulk of neuronal tau dephosphorylation [33]. Therefore, to explore whether hypoxia can result in tau hyperphosphorylation, we performed immunohistochemistry (Figure 2a,b) and Western blotting (Figure 2c,d) with specific antibodies against tau phosphorylation. Interestingly, our findings showed that chronic hypoxia induced tau hyperphosphorylation at different epitopes, including Ser199, Ser262, Ser396, and Ser404, which are common residues of tau phosphorylation regulated by GSK-3β and PP2A [27], while there was no significant difference in total tau levels between the two groups (Figure 2e). To further study the upstream events leading to tau hyperphosphorylation under hypoxia, we detected the major tau kinase GSK-3β [34] and PP2A [33]. The Western blotting data analysis showed that only the methylated PP2Ac (M-PP2Ac, the active form of PP2Ac, which is the catalytic subunit of PP2A) was significantly decreased in hypoxia rats compared with that of the control, while there were no alterations in Ser9-phosphorylated GSK-3β (the inactive form of GSK-3β), total GSK-3β, and total PP2Ac (T-PP2Ac) (Figure 2f,g). To further confirm our results, we carried out a PP2A activity test, and the data showed that hypoxia downregulates PP2A activity (Figure 2h), supporting that hypoxia-induced tau hyperphosphorylation is related to deficiency of PP2A activity. Methylesterase PME-1 and methyltransferase LCMT1 are responsible for PP2Ac methylation [35]. Therefore, we further detected the PME-1 and LCMT1 and found out that hypoxia led to a marked decrease in the level of LCMT1 but not PME-1 (Figure 2i–k). These findings suggest that chronic hypoxia might downregulate the LCMT1, then decrease the activity of PP2A, to induce tau hyperphosphorylation. The series of changes is due to the alteration of LCMT1/PP2Ac/tau phosphorylation axis.

### 2.3. Hypoxia-Induced Tau Hyperphosphorylation Positively Correlates with HIF-1α

HIF-1 is an important transcription factor that responses to alterations in cellular oxygen levels [29]. We speculated that HIF-1 might be related with the change of LCMT1/PP2Ac/tau phosphorylation axis under chronic hypoxia. HIF-1α is the functional subunit of HIF-1, so we detected the change of HIF-1α in the rats’ hippocampus. It was found that the HIF-1α expression levels both on protein (Figure 3a,b) and mRNA (Figure 3c) were significantly increased in the hippocampus of the chronic hypoxia rats compared to those of control rats. 

To further verify the upregulation of HIF-1α and the change of LCMT1/PP2Ac/tau phosphorylation axis induced by hypoxia, we treated rat primary hippocampal neurons with 1%O_2_ for continuous cultivation at different times and found that hypoxia within 48 h had no significant effect on the cell viability as suggested by the CCK8 experiment results (Figure 3d). Moreover, the levels of HIF-1α at 36 h and 48 h were significantly increased after hypoxia treatment (Figure 3e,f). Immunofluorescence assay showed a marked nuclear translocation of HIF-1α, further supporting that the levels of HIF-1a increases with hypoxia (Appendix A). Next, we detected tau phosphorylation (Figure 3g–i) and its kinase/phosphatase (Figure 3j,k) in primary neurons with hypoxia. Consistent with the experimental data derived from SD rats with chronic hypoxia treatment, 1% O_2_ treatment for 48 h in cultured neurons resulted in tau hyperphosphorylation as well as a marked decrease in M-PP2Ac, LCMT1 (Figure 3m,n), and PP2A activity (Figure 3l), while there was no change in total tau, Ser9-phosphorylated GSK-3β, total GSK-3β, T-PP2Ac and PME-1 (Figure 3m,o). Together with data from hypoxic rats, these findings suggest that HIF-1α expression is positively associated with tau hyperphosphorylation.

To further study whether HIF-1α is an upstream factor related to tau hyperphosphorylation, we employed CoCl_2_ to induce chemical hypoxia in rat C6 glioma cells stably expressing human full-length tau441 (C6/tau cells). CoCl_2_, which promotes the stabilization of HIF-1α [36,37], is commonly used as a drug for chemical hypoxia model of neurodegeneration [38]. Global hypoxia in animals can affect the expression of a variety of substances, and the chemical hypoxia induced by CoCl_2_ specifically changes HIF-1α. Firstly, we tested the optimal concentration of CoCl_2_ on C6/tau cells. After CoCl_2_ treatment for 8 h, there was no significant alteration in the cell survival rate with the increase of CoCl_2_ concentration till 1200 µmol/L (Appendix A). Based on the cell viability experiment results, 50 μmol/L and 100 μmol/L CoCl_2_ showed less effect on cell viability and thus were chosen for subsequent experiments in this study. As expected, the CoCl_2_ treatment significantly increased the expression level of HIF-1α (Appendix A). In addition, the Western blot results also showed that chemical hypoxia by CoCl_2_ led to tau hyperphosphorylation at Ser262 and Ser 404 without alteration of total tau (Appendix A), a decrease in M-PP2Ac and LCMT1 levels, but no changes in T-PP2Ac and PME-1 levels (Appendix A). Taken together, our data suggests that hypoxia downregulates LCMT1/PP2Ac and promotes tau hyperphosphorylation, maybe by upregulating the level of HIF-1α.

### 2.4. Downregulation of HIF-1 Prevents Hypoxia-Induced Alterations in the LCMT1/PP2Ac/Tau Phosphorylation Axis in Primary Hippocampal Neurons

To clarify the HIF-1α involvement in hypoxia-induced LCMT1/PP2A/tau dysfunction, we constructed lentivirus siHIF-1α and evaluated the effect of HIF-1α deficiency on the LCMT1/PP2Ac/tau phosphorylation axis under hypoxia condition in primary hippocampal neurons. Firstly, we constructed three siHIF-1α plasmids, and chose the plasmid (6315) with the highest efficiency for virus packaging (Appendix A). Next, we examined the efficiency of lentivirus siHIF-1α to downregulate HIF-1α and found out that the HIF-1α level in siHIF-1α-treated hypoxia group was significantly decreased compared to control and hypoxia groups without siHIF-1α treatment (Figure 4a,b). Interestingly, the results from the Western blotting showed that siHIF-1α attenuated hypoxia-induced tau hyperphosphorylation and had no effect on the level of total tau (Figure 4a,c,d). Intriguingly, downregulation of HIF-1α recovered the levels of the decreased PP2Ac methylation (Figure 4e,f) and LCMT1 (Figure 4e,h), while it had no effect on the total PP2Ac and PME-1 (Figure 4e,g,i). These results suggest that HIF-1α is a key factor and is required for the series changes of LCMT1/PP2Ac insult and tau hyperphosphorylation during chronic hypoxia.

To further verify the effect of HIF-1α on tau phosphorylation, PP2Ac methylation and LCMT1, 2ME2 was employed to inhibit the function of HIF-1α [39]. 50 µmol/L 2ME2 was used to treat the primary hippocampal neurons for 24 h (see Section 4). The result showed that hypoxia treatment accompanied with 50 µmol/L 2ME2 dramatically suppressed the increased expression of HIF-1α induced by hypoxia (Figure 4j,k). It also suppressed tau phosphorylated at pS396 induced by hypoxia, without changing the total tau (Figure 4j,l). Inhibitor HIF-1α by 2ME2 increased the level of M-PP2Ac (Figure 4 j,m) and LCMT1 (Figure 4j,n). These results further demonstrated the important role of HIF-1α in tau phosphorylation, and the series changes of LCMT1/PP2Ac. 

### 2.5. Knockdown of HIF-1α Relieves the Cognitive Dysfunction Induced by Chronic Hypoxia in Rats

In AD, hyperphosphorylation of tau is associated with cognitive impairments [40]. To further investigate whether downregulation of HIF-1α attenuates the disorder of the LCMT1/PP2Ac/tau phosphorylation axis and successively rescues cognitive insults caused by chronic hypoxia, we constructed AAV9-siHIF-1α and examined the effect of HIF-1α deficiency on the behavior and tau pathology under hypoxia condition in SD rats. In the present study, sixty healthy SD rats were randomly divided into control (AAV-vector + normoxia), hypoxia (AAV-vector + hypoxia), siHIF-1α+hypoxia, and siHIF-1α+ normoxia groups. Figure 5a shows the experimental procedure. We first examined the efficiency of AAV9-siHIF-1α and found out that the HIF-1α level in the siHIF-1α-treated rats were significantly decreased compared to that of the hypoxia groups (Figure 5b,c). Then, we carried out OFT and found that there was no significant difference in the total distance covered among the groups (Figure 5d), suggesting that HIF-1α has no effect on locomotor functions. The NOR results showed that siHIF-1α recovered the decreased exploration time for new objects caused by chronic hypoxia (Figure 5e), indicating that knockdown of HIF-1α restores the curiosity toward exploring new things in these rats. The MWM test and fear conditioning test revealed that downregulation of HIF-1α rescued the chronic hypoxia-induced spatial learning/memory loss (Figure 5f–i) and the impaired fear memory (Figure 5j). Together, these data demonstrate that knockdown of HIF-1α restores learning and memory impairments in chronic hypoxia rats, further suggesting the implication of HIF-1α in AD-like pathology following chronic hypoxia.

### 2.6. Downregulation of HIF-1α Recovers LCMT1/PP2Ac/tau Axis Dysfunction in Rats’ Hippocampus

Knockdown of HIF-1α avoided the cognitive dysfunction induced by chronic hypoxia in rats, so next we tested the effect of downregulated HIF-1α on the change of LCMT1/PP2Ac/tau phosphorylation axis. Both immunohistochemical assays (Figure 6a,b) and Western blotting (Figure 6c–e) showed that siHIF-1α attenuated hypoxia-induced tau hyperphosphorylation in the hippocampus. Moreover, downregulation of HIF-1α recovered the decreased levels of PP2Ac methylation (Figure 6f,g) and LCMT1 (Figure 6f,i) and increased PP2A enzyme activity (Figure 6k), while no effect was observed on the levels of total PP2Ac and PME-1 (Figure 6f,h,j). These findings implied that HIF-1α causes LCMT1/PP2Ac deficiency and tau hyperphosphorylation, mediating the chronic hypoxia-induced cognitive deficits.

### 2.7. HIF-1α Accelerates LCMT1 Degradation, Counteracting its Transcriptional Increase

Based on the above findings that LCMT1 was decreased under hypoxia conditions and the fact that HIF-1α was induced by hypoxia, we speculate that HIF-1α may be a transcription factor of LCMT1. To investigate this, we employed a dual luciferase assay and the results showed that HIF-1α binds to LCMT1 (Figure 7a). Our data above showed that HIF-1α was negatively associated with the protein expression of LCMT1, however, overexpression of HIF-1α unexpectedly promoted the expression of LCMT1 mRNA (Figure 7b), suggesting that HIF-1α positively regulates LCMT1 at the transcriptional level. To confirm the effect of HIF-1α on the protein level of LCMT1, we transfected C6 cells with HIF-1α plasmid and performed Western blotting. The protein level of HIF-1α was significantly increased following HIF-1α plasmid transfection (Figure 7c,d), and the protein level of LCMT1 was markedly decreased when compared with the control group (Figure 7c,e). Therefore, we suspect that overexpression of HIF-1α might accelerate LCMT1 degradation. To address this, we transiently transfected the C6 cells with or without HIF-1α for 24 h and then treated the cells with cycloheximide (CHX), an inhibitor of protein biosynthesis, at different time points. We found that overexpression of HIF-1α obviously accelerates LCMT1 degradation (Figure 7f,g). Together, these data suggest that HIF-1α speeds up LCMT1 degradation which counteracts the LCMT1 transcriptional expression, resulting in an increased mRNA but a decreased protein level of LCMT1.

To understand how HIF-1α induces LCMT1 degradation, we treated primary hippocampal neurons with CoCl_2_ to induce chemical hypoxia. CCK8 assay revealed no toxic effect of CoCl_2_ on cell activity at 25 μmol/L (Figure 8a). This concentration was therefore used in the subsequent experiments. Next, we respectively employed chloroquine (CQ, a lysosome inhibitor) and MG132 (proteasome inhibitor) to inhibit the autolysosome pathway and the ubiquitin-proteasome pathway and assessed the LCMT1 level after CoCl_2_ treatment. Interestingly, only CQ but not MG132 blocked the decreased LCMT1 level caused by CoCl_2_ (Figure 8b–e), suggesting that the autophagy-lysosome system is associated with HIF-1α-induced LCMT1 degradation. To further investigate the above results, we used CHX and examined the level of HIF-1α, LCMT1, and the autophagy-related molecule LC3 after CoCl_2_ treatment. Western blotting showed that CoCl_2_ induced an increase in HIF-1α and a decrease in LCMT1. Importantly, LC3-II/LC3-I was also significantly increased following CoCl_2_ treatment as compared to control (Figure 8f–i), implying that HIF-1α promotes autophagy. It has been previously reported that LCMT1 is cleared in the late endosomes [41], thus we detected the LCMT1 distribution in late endosomes. The immunofluorescence results revealed an increased colocalization of LCMT1 and RAB7 (late endosome marker) in neurons treated with CoCl_2_ as compared to control (Figure 8j,k). These data strongly support that HIF-1α promotes autophagy and accelerates LCMT1 clearance in late endosomes.

Taken together, our findings suggest that chronic hypoxia upregulates HIF-1α, which obviously accelerates LCMT1 degradation counteracting its transcriptional expression. This leads to a decrease in LCMT1 protein level which in turn decreases PP2Ac methylation and thus activity, finally resulting in tau hyperphosphorylation and cognitive impairment.

## 3. Discussion

Different environmental changes and physiological changes can lead to hypoxia. In our previous research, we found that acute anoxia, but not glucose deprivation, induces tau dephosphorylation at the PHF-1, Tau-1, R145 and AT270 epitopes, and the activation of PP-2A, probably through its dephosphorylation at Tyr307, may play a major role in acute anoxia-induced tau dephosphorylation [42]. Chronic hypoxia is reported as an important environmental factor contributing to the development of sporadic AD with two typical neuropathologic alterations Aβ pathology and tau hyperphosphorylation [30,43,44]. Accumulating data have clarified that chronic hypoxia upregulates HIF-1α, mediating Aβ toxicity by promoting Aβ generation and decreasing Aβ clearance [21,22,45]. However, until now the exact molecular mechanism underlying chronic hypoxia-related tau pathology, the other hallmark of AD, remains ambiguous. In the current study, we provide extensive evidence demonstrating chronic hypoxia upregulating HIF-1α, decreasing the level of LCMT1, leading to PP2A deficiency, and resulting in tau hyperphosphorylation and cognitive impairments. Moreover, we exhibit additional evidence that blocking HIF-1α expression reverses the decreased LCMT1 level and PP2A activity, tau hyperphosphorylation, and cognitive defects caused by chronic hypoxia. Meanwhile, we proved that HIF-1α obviously accelerates the autophagy-lysosomal pathway-related LCMT1 degradation counteracting LCMT1 transcriptional expression, thus resulting in a marked decrease in LCMT1 level during chronic hypoxia. Together, our data strongly support the notion that HIF-1α regulates LCMT1 and lowers PP2A activity, mediating tau phosphorylation and cognitive impairments in chronic hypoxia.

In the clinic, pulmonary diseases such as obstructive sleep apnea syndrome and chronic obstructive pulmonary disease can significantly decrease oxygen supply and therefore can cause chronic hypoxia, and have been reported to be closely associated with cognitive defects [46,47,48]. In the present study, we induce global hypoxia (10% O**_2_**, 6 h per day) for one month to imitate a chronic hypoxia condition in SD rats. Although open field test (OFT) results showed that hypoxia rats displayed similar patterns of anxiety and exploratory activity as control animals, the novelty recognition experiment (NOR) and the Morris water maze (MWM) showed that chronic hypoxia caused learning and memory impairments in rats. Since hyperphosphorylation of tau is related to cognitive dysfunction [49,50], we evaluated the phosphorylation status of tau, and consistent with previous studies [30,43], we found that chronic hypoxia caused tau hyperphosphorylation in both the hippocampus of rats and in cell cultures.

It is well known that tau hyperphosphorylation is regulated by its kinase and phosphatase [24,51]. GSK-3β, known as human tau protein kinase [52] is one of the main kinases that phosphorylate tau at different epitopes. Phosphorylation of GSK-3β is regarded as the main regulatory mechanism of its activity in physiological and some pathological conditions. GSK-3β phosphorylation at Ser9 is associated with its inhibition and thus inactivity [53,54,55]. We found that chronic hypoxia did not affect the levels of total GSK-3β or its Ser9 phosphorylation, indicating that GSK-3β isn’t involved in chronic hypoxia-caused tau hyperphosphorylation. Conversely, protein PP2A, known as a major regulator of tau phosphorylation [33,56], showed a significant decrease in both the activity of its catalytic subunit (PP2Ac) and its methylated (activated) form at leucine 309 (PP2Ac-L309) [57] in chronic hypoxia rats when compared to control, indicating that chronic hypoxia inhibits PP2A activity via downregulation of PP2Ac methylation and subsequently leads to tau hyperphosphorylation. PP2Ac methylation is regulated by Methylesterase PME-1 and methyltransferase LCMT1 [35]. We observed a marked decrease in the protein level of LCMT1 but not PME-1 in hypoxia-treated rats and cells. These findings strongly imply that chronic hypoxia may downregulate the LCMT1/PP2Ac axis and hence induce tau hyperphosphorylation. 

HIF-1α, a transcription factor responsive to hypoxia, has been reported to be associated with AD [45,58,59]. However, the exact role of HIF-1α in tau pathology remains controversial. Liu’s study showed that downregulation of HIF-1α caused GLUT1 and GLUT3 deficiency, which was associated with the decrease in O-GlcNAcylation, tau hyperphosphorylation, and the density of neurofibrillary tangles in AD brain [60]. On the other hand, Shin N et al. reported that repetitive mild traumatic brain injury induces hypoxia and increases HIF-1α, resulting in tau hyperphosphorylation in neurons and astrocytes in both the hippocampus and cortex [61]. In the present study, we found that chronic hypoxia-induced tau hyperphosphorylation positively correlates with HIF-1α. Moreover, increased HIF-1α caused by CoCl_2_ induces LCMT1/PP2Ac deficiency and tau hyperphosphorylation. Our further study showed that conversely, HIF-1α silencing recovered chronic hypoxia-induced LCMT1/PP2Ac/tau axis alterations and attenuated cognitive impairment, strongly supporting that HIF-1α downregulates the LCMT1/PP2Ac axis to induce tau pathology.

We proved that HIF-1α functions as a transcription factor of LCMT1 as evidenced by the dual luciferase assay, but unexpectedly, HIF-1α negatively regulates the expression of LCMT1, hinting at an increased degradation that counteracts the HIF-1α upregulation of LCMT1 expression. Interestingly, the degradation pathway studies indicate that only CQ (chloroquine) but not MG132 blocked the decreased LCMT1 level caused by chronic hypoxia, suggesting that the autophagy-lysosome system but not the proteasomal system is associated with HIF-1α-induced LCMT1 degradation. This is further supported by the immunofluorescence staining results which show increased fluorescence intensity of LCMT1 in the late endosomes.

## 4. Materials and Methods

### 4.1. Plasmids, Viruses, Chemicals, and Antibodies

The detailed information of the plasmids and viruses is found in the Appendix A. In brief, the plasmids of HIF-1α (NM_024359, GV712- XbaI/XbaI- HIF-1α), LCMT1 (NM_199405, GV238- KpnI/XhoI-LCMT1), LV-shHIF-1 (hU6-MCS-CBh-gcGFP-IRES-puromycin), AAV9-Vector (U6-MCS-CAG-EGFP), and AAV9-siHIF-1α (U6-MCS-CAG-EGFP) were constructed and packaged by Genechem Co., Ltd (Shanghai, China). The plasmids encoding pCMV-MCS-SV40-puromycin (tau441) were generated in our laboratory [62]. All plasmids were sequenced and prepared using an endotoxin-free plasmid extraction kit (Tiangen biotech Co., Ltd., Beijing, China). HighGene transfection reagents were from ABclonal Technology Co., Ltd (Wuhan, China, RM09014). Reagents for cell culture were from Gibco BRL (Gaithersburg, MD, USA). The primary antibodies used in this study and their properties are shown in Table 1. Secondary antibodies for Western blotting were purchased from Amersham Pharmacia Biotech (Little Chalfort, Buckinghamshire, UK). Dyed with DAB (Zsbio Commerce Store, Beijing, China, ZLI-9017) and CoCl_2_ was from Sigma (St. Louis, MO, USA, #255599). 

### 4.2. Animals

All rats used in this study were male Sprague-Dawley (SD) rats (2 months old, 250 ± 20 g) supplied by the Experimental Animal Center of Tongji Medical College, Huazhong University of Science and Technology. Rats were kept under standard laboratory conditions: 12 h alternating light/dark cycle with water and food ad libitum. Adapting to the environment for one week, the rats were randomly divided into different groups and treated as stated in different parts of the study. All animal experiments were approved by the Animal Care and Use Committee of Huazhong University of Science and Technology.

### 4.3. Chronic Hypoxia in Animals

The animal chronic hypoxic model was performed in accordance with a classical method [63]. The rats were placed in the incubator, and the door of the incubator was closed with plastic wrap. After a certain amount of nitrogen was filled into the incubator, the oxygen concentration in the incubator was detected with an oxygen meter. The oxygen concentration was controlled at 10% for hypoxia. It continued for one month, and 6 h at a fixed time (9:00 am–3:00 pm) every day. The other group was kept in a normal O_2_ concentration (21% O_2_) in the same room as normoxic controls.

### 4.4. Lateral Ventricle Stereotactic Injection

The rats were anesthetized with 1% pentobarbital sodium (35 mg/kg) and then stabilized on a stereotaxic instrument (RWD, Shenzhen, China). After being sterilized with iodophors and 75% (*vol*/*vol*) ethanol, the scalp was incised along the skull midline. Two holes were stereotaxically drilled in the bilateral skull at posterior 1.2 mm, lateral 2.6 mm, and depth 4.0 mm from the bregma. Using a microinjection system (World Precision Instruments, Sarasota, FL, USA), AAV9-vector/AAV9-siHIF-1α was injected into the bi-lateral ventricle (2.5 µL for per injection site) at a rate of 0.25 µL/min before chronic hypoxia. The needle syringe was kept in place for 10 min before withdrawal, and then the skin was sutured and sterilized with iodophors and 75% ethanol. 

### 4.5. Animal Behavior Tests

#### 4.5.1. Open Field Test

The open field test (OFT) is used to assess anxiety and exploration activities, as described earlier [64]. The test equipment is a typical open field (100 × 100 cm^2^ PVC square arena with 70 cm high walls), supplied by Shanghai Xinruan Technology Co, LTD. Rats were individually trained for a single 5 min period. Anxiety is studied by analyzing the percentage of time spent in the middle of the arena. The total distance, average speed, center duration%, and center distance% were used to assess the general locomotor function of the rats.

#### 4.5.2. Novel Object Recognition

Novel object recognition (NOR) is a learning and memory test method established using the principle of animal innate tendency to explore new objects. The rats were placed in the arena (100 cm × 100 cm × 70 cm container) for 5 min to get familiar with the arena. The day after, the rats reentered the arena from the same starting point and were granted 5 min to familiarize themselves with object A and object B. Exactly 2 h after the familiarization period, object B (familiar object) was replaced with object D (new object), and the rats were granted another 5 min to explore both objects. After 24 h, object D (now, the familiar object) was replaced with object C (now, the new object), and the rats were granted again 5 min to explore both objects. The behavior was recorded by a video camera positioned above the arena. The recognition index was calculated as TA/(TA + TC), TC/(TA + TC), TA/(TA + TD), and TD/(TA + TD). TA, TB, TC, and TD were the time rats explored the objects A, B, C, and D, respectively.

#### 4.5.3. Morris Water Maze Test

Spatial learning and memory were detected by the Morris water maze (MWM) test. For spatial learning, the rats were trained in the water maze to find a hidden platform for 5 days, from 9:00 am to 3:00 pm. Rats were trained to be able to find the submerged platform within 60 s in each trial. If the rats failed to find the hidden platform within 60 s, they were manually guided to the platform and stayed there for 20 s. Each rat was subjected to 15 training trials: 3 trials per day for 5 training days. For the spatial memory test on day 6, the platform was removed, and the latency for the first crossing of the position of the platform, platform crossings times, duration in platform quadrant, and non-platform quadrants were monitored by the Noldus video tracking system (Ethovision, Noldus Information Technology, Wageningen, The Netherlands).

#### 4.5.4. Fear Conditioning Test

A fear conditioning test is intended to elucidate a subject’s ability to associate a conditioned stimulus with an aversive, unconditioned stimulus, such as foot shock. The test included two periods. The first period involved training: rats were placed in the chamber, and after 3 min a sound stimulus was administered for 10 s. Then, a short-term current stimulation (0.8 mA, 3 s) immediately followed. The current stimulation cycle was repeated three times. The second period is the test: this period takes place either 2 h or 24 h following the training and involved only sound stimulation, with no current administered. The freezing times, as the rats await the short-term current stimulus, were then recorded.

#### 4.5.5. Primary Neuron Culture and Treatment

Primary cultures of rat hippocampal neurons were prepared from E18 Sprague-Dawley rat embryos as previously reported [65]. Briefly, the hippocampi from the pregnant rat pups were dissected and gently chopped in Hank’s buffered saline solution and then suspended in a 0.25% (*v*/*v*) trypsin solution for 15 min at 37 °C. Neurons were plated in 6-well and 12-well plates coated with 100 μg/mL poly-D-lysine and then cultured in a Neurobasal medium supplemented with 2% (*v*/*v*) B-27 and 1× Glutamax. Half of the media was changed every 3 days. Hippocampal neurons were cultured at 37 °C in a humidified 5% (*v*/*v*) CO_2_ incubator. The neurons were cultured for 7 days and then used for different experiments.

For the hypoxia experiment, the cells were cultured with 1% O_2_. Meanwhile, the control cells were cultured with normoxia. For knockdown of HIF-1α in primary cultures of neonatal SD rat neurons, the cells were infected by lentiviral vectors carrying short hairpin RNA (shRNA) targeting HIF-1α. This was followed 4 days later by hypoxic exposure, while the control neurons were infected with the empty virus only with or without subsequent hypoxic exposure. For chemical hypoxia, the primary neurons were treated with different concentrations of CoCl_2_ for 8 h.

For the HIF-1α inhibiting experiment, primary neurons were culture and treatment with the inhibitor 2ME2 (GLPBIO, Montclair, CA, USA) for 24 h. 2ME2 was constituted in DMSO as a stock solution, and further diluted in the culture medium to a final concentration of 50 µmol/L.DMSO was diluted in the same manner to a final concentration of 0.5% as the vehicle control [66].

#### 4.5.6. C6 Cell Culture, Plasmid Transfection, and Treatment

Rat C6 glioma cells (C6 cells) were cultured in Dulbecco’s modified Eagle’s medium (Gibco) containing 10% fetal bovine serum Gibco BRL, Gaithersburg, MD, USA), in a humidified aerated incubator with 5% CO_2_ at 37 °C. Cells were seeded into 6-well cultured plates and incubated for 24 h prior to transient transfection. Each well was transfected with the mixture containing a total of 3 μg tau441, HIF-1α, or LCMT1 plasmid and 6 μL HighGene transfection reagents (ABclonal) according to the manufacturer’s protocols. For chemical hypoxia, the cells were treated with different concentrations of CoCl_2_ for different periods according to the experimental requirements.

#### 4.5.7. Cell Counting Kit 8 Assay

The cell viability in different groups was measured using a Cell Counting Kit 8 assay (CCK-8 assay). Cells were cultured in 96-well plates (100 μL, 5 × 103 per well). The cells were treated with different hypoxia treatments, and then CCK-8 reagent (ab228554, Abcam, Cambridge, MA, USA) and DMEM were added to each well. After incubation with the CCK-8 reagent, the optical density of cells was measured at 450 nm at different time points using a microplate reader (Bio-Tek, Winooski, VT, USA).

### 4.6. Western Blotting

Cells samples were incubated with a buffer containing RIPA low lysis buffer and protease inhibitor on ice for 10 min and then collected. For brain tissue, the rats’ hippocampi were rapidly removed and were homogenized at 4 °C using a buffer containing 50 mmol/L Tris-HCl, pH 7.4, 150 mmol/L NaCl, 10 mmol/L NaF, 1 mmol/L Na3VO4, 5 mmol/L EDTA, 2 mmol/L benzamidine, and 1 mM PMSF. After centrifugation of the tissue homogenates at 12,000 rpm/min, the supernatants were collected. Protein concentrations were quantified by a Bicinchoninic acid (BCA) protein kit (Pierce, Rockford, IL, USA), and β-mercaptoethanol (BME), and bromophenol blue (3:1) was added. The proteins in the extracts were separated by 10% SDS-PAGE and transferred to a nitrocellulose membrane. The membranes were first blocked and then incubated with primary antibodies (see Table 1) overnight at 4 °C. Then, after three washes with TBST, the membranes were incubated with the secondary antibody at room temperature for 1 h. An Odyssey Infrared Imaging System was used to visualize immunoreactive bands. Bands’ intensity readings were obtained using ImageJ software. Western blotting of the cells was performed in three independent experiments. Each band was individually framed and quantified for analysis.

### 4.7. RNA Extraction and Real-Time PCR

Total RNA from the cells was extracted by the Trizon Reagent method (CWBIO, Wuhan, China). First-strand complementary DNA (cDNA) was synthesized from a total of 50 mg RNA using a ReverTra Ace qPCR RT kit (TOYOBO, New York, NY, USA). Quantitative polymerase chain reaction (PCR) was performed in a standard PCR reaction mixture prepared in duplicate using an Applied Biosystems 7900 Prism Real-Time PCR system and SYBR Premix Ex Taq (TaKaRa, Dalian, Japan) in accordance with the manufacturer’s protocol. Quantitative PCR primers were shown in Table 2.

### 4.8. PP2A Activity Assay

A Serine/Threonine Phosphatase Assay System kit V2460 was used to detect PP2A activity in the primary neurons and brain homogenates according to the manufacturer’s procedure (Promega, Madison, WI, USA). In brief, cells or brain tissues were homogenized at 4 °C for 30 min using a phosphatase-free storage buffer (10,000 cells: 1 μL or 1 g tissue: 3 mL). Then, cell lysates or tissue extracts were loaded into a spin column and centrifuged at 600× *g* at 4 °C for 5 min to remove the free phosphates. Protein concentrations of the phosphate-free samples were then assayed by using a BCA kit (Pierce, Rockford, IL, USA). Protein samples of 5μg were incubated in triplicates with a chemically synthesized phosphopeptide (RRA (pT) VA), an optimal substrate for PP2A, PP2B, and PP2C, but not for PP1, in a buffer optimized for PP2A activity while cation-dependent PP2B and PP2C were inhibited (protocol provided by the manufacturer) for 30 min at 32 °C. Phosphate released from the substrate was detected by measuring the absorbance of a molybdate-malachite-green–phosphate complex at 630 nm. PP2A activity was evaluated by the release of phosphate per μg protein and per minute (p moL/μg/min).

### 4.9. Immunohistochemistry

The rats were anesthetized and transcardially perfused with normal saline followed by 4% paraformaldehyde. The dissected brains were post-fixed for 48 h and then cryoprotected with 30% sucrose and subsequently in optimum cutting temperature (OCT) compound for cryostat sectioning. The brains were coronally sliced with a 30 μm thickness by a freezing microtome (CM1900, Leica, Wetzlar, Germany). The brain slices were mounted on gelatin-coated slides. After rinsing with 0.1% Triton in PBS, endogenous peroxidase was quenched with PBS containing 3% H_2_O_2_ for 15 min, and then the slices were blocked in PBS buffer containing 5% goat serum albumin and 0.1% TritonX-100 for 30 min. The sections were then incubated with specific primary antibodies (Table 1) at 4 °C overnight. After being rinsed with PBS, the sections were then incubated with secondary fluorescence or biotinylated antibodies (Abbkine, HRP, Goat Anti-Rabbit lgG, Wuhan, China, 1:200). The immunoreactions were developed using a DAB-staining kit (Zsbio Commerce Store, Beijing, China). Sections were then dehydrated through graded ethanol series, and sealed with neutral balsam. Images were obtained with an Olympus BX61 microscope and analyzed with Fiji software (ImageJ).

### 4.10. Immunofluorescence

The cells were cultured on coverslips and fixed in 4% (*v*/*v*) paraformaldehyde for 15 min and then permeabilized in PBS containing 0.5% Triton X-100. Non-specific binding was blocked by incubating in PBS containing 0.1% Triton X-100 and 5% BSA for 30 min. The primary antibodies were then applied diluted in the blocking solution and incubated at 4 °C overnight. The secondary antibodies (Jackson immune research, West Grove, PA, USA 1:200) were added to the coverslip for 1 h at room temperature. The coverslips were washed and mounted onto slides. The images were observed with a laser confocal microscope (710; Zeiss, Oberkochen, Germany).

### 4.11. Targeting Relationship Verification by Dual-Luciferase Reporter Gene Experiment

A luciferase assay was performed as described previously [67]. The binding sites of HIF-1α and LCMT1 were predicted by Jaspar online tools. Logarithmic growth phase C6 cells were cultured in a 6-well plate, and cells with a confluence of 80% were then transfected. After transfection, cells were continuously incubated for 48 h, and each well was added with 100 µL of lysate. Following that, centrifugation was carried out, and a 96-well plate was used to collect the supernatant. Then 40 µL of firefly luciferase substrate was added to each well, and following gentle mixing for 10 s, the fluorescence intensity was measured. For reference, 40 µL of Renilla luciferase substrate was also added to the 96-well plate; then, the luciferase activity was measured with a Glomax luminometer (Promega, Madison, WI, USA).

### 4.12. Statistical Analyses

All data were expressed as mean ± SD and analyzed by GraphPad Prism 6 statistical software (GraphPad Software, Inc., La Jolla, CA, USA). The one-way analysis of variance (ANOVA) procedure was used to determine the differences among groups followed by LSD’s post hoc tests to determine the differences among two groups. *p* < 0.05 was considered statistically significant in all experiments. All results shown corresponded to individual representative experiments.

## 5. Conclusions

Our data strongly support that chronic hypoxia promotes HIF-1α expression, leading to LCMT1/PP2Ac deficiency and tau hyperphosphorylation, finally resulting in cognitive impairments. Given the deleterious effects that the disorder of the HIF-1α/LCMT1/PP2Ac/tau axis in AD exerts on numerous pathological events, we speculated, investigated, and provided evidence that blockage of HIF-1α or HIF-1α/LCMT1 might provide pharmacological interference for chronic hypoxia-related tau pathology.

## Figures and Tables

**Figure 1 ijms-23-16140-f001:**
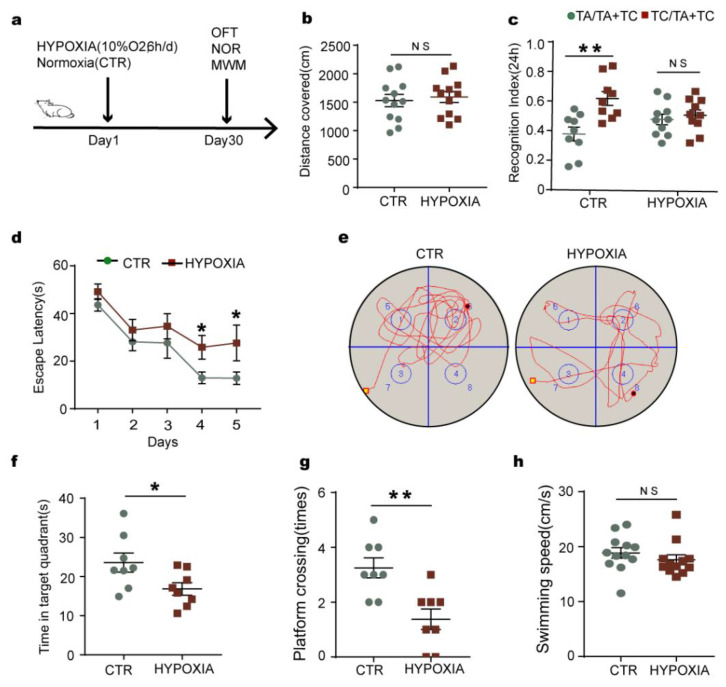
Chronic hypoxia impairs cognitive functions in rats. (**a**) Experimental design. (**b**) The open field test (OFT) showed no difference in the total distance covered between the two groups. (**c**) The novel object recognition (NOR) test showed the measured recognition index of the new object within 24 h. (**d**–**h**) In the Morris water maze (MWM) test, the latency to find the hidden platform from day 1–5 (**d**); on day 6, spatial memory was tested by removing the platform (**e**). The time spent in the target quadrant (**f**), platform crossing times (**g**), and swimming speed at the day 6 (**h**) were measured. n = 8–12. All data are presented as mean ± SD. * *p* < 0.05, ** *p* < 0.01 vs. control group.

**Figure 2 ijms-23-16140-f002:**
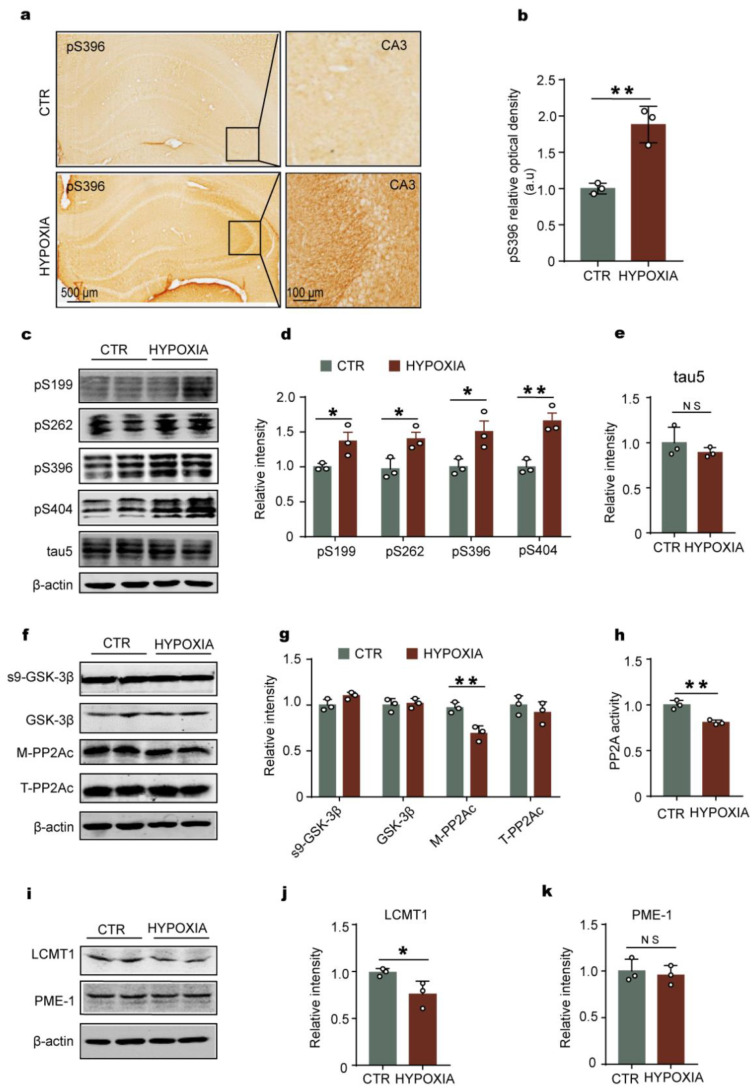
Chronic hypoxia leads to tau hyperphosphorylation, accompanied by loss of LCMT1-related PP2A activity in the hippocampus of rats. (**a**,**b**) Immunohistochemistry showed an increase in pS396 positive staining in hypoxic rats compared to control (scale bars = 500 μm and 100 μm). (**c**–**e**) Hippocampal tissues were homogenized, and phosphor-tau protein levels at pS199, pS262, pS396, and pS404 sites were detected by immunoblotting (**c**,**d**). Total tau level was measured (**e**) with actin as the loading control. (**f**,**g**) The levels of the total GSK-3β and the Ser9- phosphorylated GSK-3β (s9-GSK-3β), Leu309-methylated (M-PP2Ac), and total PP2Ac (T-PP2Ac) were measured by Western blotting. (**h**) PP2A activity assay in different groups in rat hippocampus. (**i**–**k**) The levels of PP2Ac-specific PME-1 and LCMT1 were measured by Western blotting. All data are presented as mean ± SD. *n* = 3 (rats per group). * *p* < 0.05, ** *p* < 0.01 vs. control.

**Figure 3 ijms-23-16140-f003:**
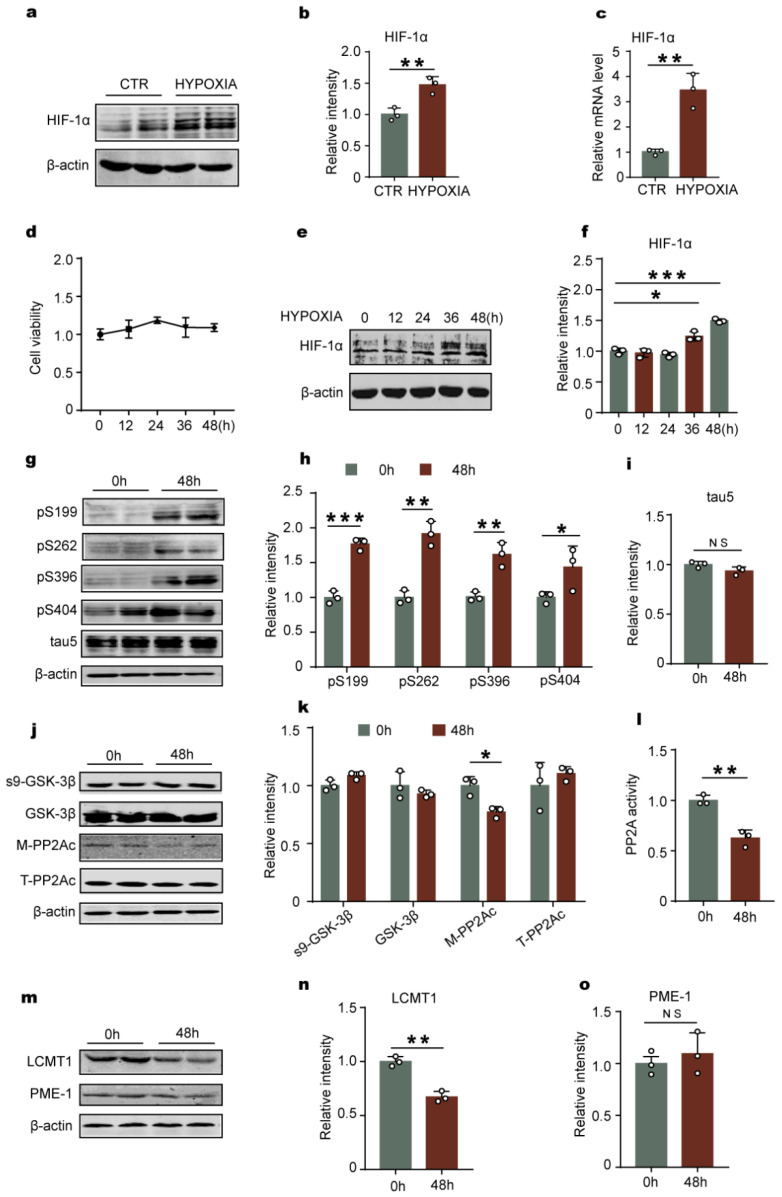
Hypoxia upregulates HIF-1α and leads to tau hyperphosphorylation in rat primary hippocampal neurons. (**a**,**b**) HIF-1α protein level was significantly higher in the hippocampus of hypoxia group compared with that of the control group as evaluated by western blotting, *n* = 3 rats per group. (**c**) HIF-1α mRNA expression level of hypoxia group was also increased, *n* = 3 rats per group. (**d**) CCK8 assay in primary neurons subjected to hypoxia for different times (0 h, 12 h, 24 h, 36 h, 48 h). (**e**) Western blots for HIF-1α in primary neurons. (**f**) Quantification of the HIF-1α protein expression levels after normalization to the β-actin signal. (**g**) Western blots for tau phosphorylation levels at pS199, pS262, pS396, pS404 sites, and total tau (Tau5) in neurons. (**h**,**i**) Quantification of the relative protein expression levels of pS199, pS262, pS396, pS404 tau, and Tau-5 normalization to the β-actin signal. (**j**,**k**) Western blots and quantitative analysis of S9-GSK3β, GSK3β, the catalytic subunit of PP2A (m-PP2Ac), and total PP2Ac in neurons. (**l**) PP2A activity assay in different groups in neurons. (**m**–**o**) Western blots and quantitative analysis of LCMT1 and PME-1 in neurons. All data represent mean ± SD, *n* = 3, * *p* < 0.05, ** *p* < 0.01 and < 0.001 vs. control group.

**Figure 4 ijms-23-16140-f004:**
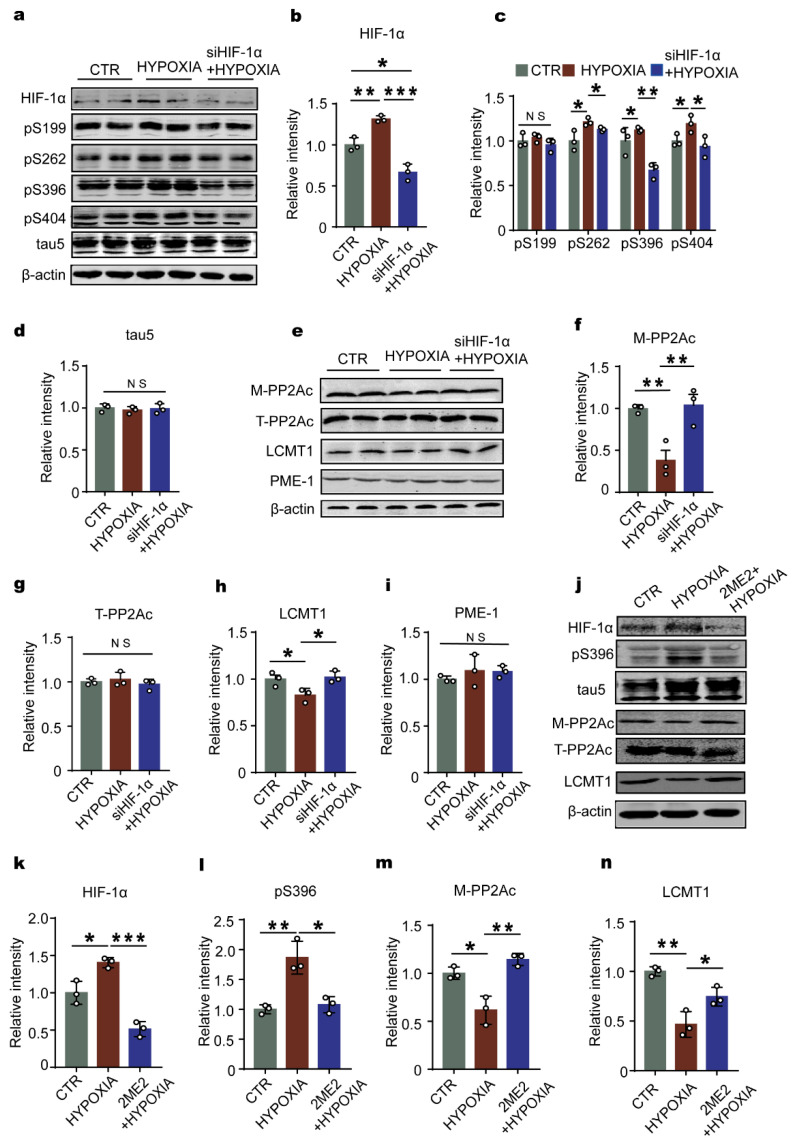
Inhibition of HIF-1α expression reduces tau phosphorylation levels and reactivates PP2A in primary neurons. (**a**) Western blots for HIF-1α and tau phosphorylation levels at pS199, pS262, pS396, pS404 sites, and tau5 in primary neurons. (**b**–**d**) Quantification of the relative protein expression levels (HIF-1α, pS199, pS262, pS396, pS404, and tau5) after normalization to the β-actin signal. (**e**–**i**) Western blots and quantitative analysis of T-PP2Ac, M-PP2Ac, LCMT1 and PME-1 in the primary neurons. (**j**) Western blots for HIF-1α and tau phosphorylation levels at pS396, tau5, M-PP2Ac, T-PP2A and LCMT1 in primary neurons. (**k**–**n**) Quantification of the relative protein expression levels (HIF-1α, pS396, M-PP2Ac and LCMT1) after normalization to the β-actin signal. Data represent mean ± SD, *n* = 3, * *p* < 0.05, ** *p* < 0.01, *** *p* < 0.001 vs. control.

**Figure 5 ijms-23-16140-f005:**
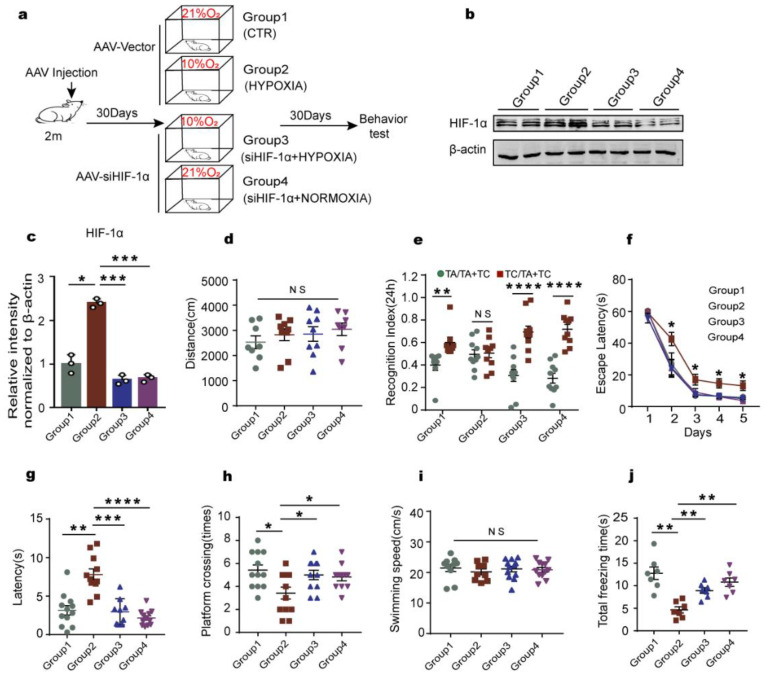
Downregulation of HIF-1α rescues chronic hypoxia-induced cognitive impairments in rats. (**a**) Study design timeline and grouping: control (Group 1), hypoxia (Group 2), siHIF-1α+hypoxia (Group 3), and siHIF-1α+normoxia (Group 4). (**b**) Western blots for HIF-1α protein level in groups. *n* = 3 rats per group. (**c**) Quantification of the relative protein expression levels of HIF-1α after normalization to the β-actin signal. (**d**) The total distance covered in the OFT. (**e**) NOR showed the measured recognition index of the new object within 24 h. (**f**–**i**) The result of MWM test: the latency to find the hidden platform from day 1–5 (**f**), latency to first cross the position of the platform on test day 6 (**g**), platform crossing times (**h**), and swimming speed at day 6 (**i**). (**j**) In the fear memory test, the total freezing time was analyzed on test day. n = 7–12. All data are presented as mean ± SD. * *p* < 0.05, ** *p* < 0.01, *** *p* < 0.001 and **** *p* < 0.0001 vs. control group.

**Figure 6 ijms-23-16140-f006:**
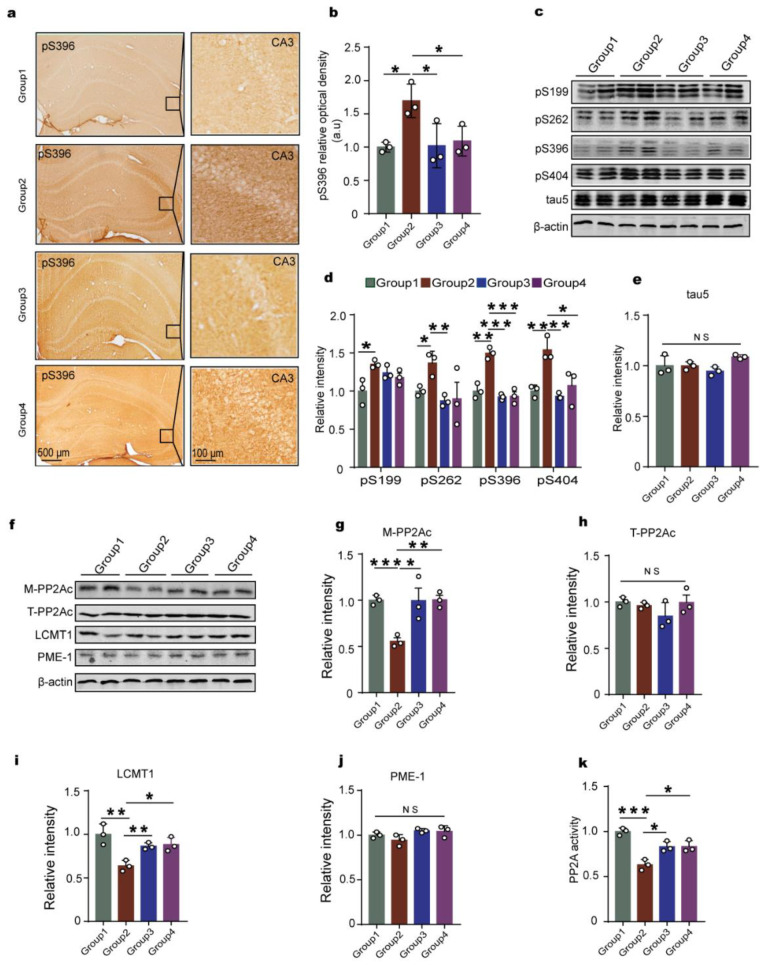
Downregulation of HIF-1α blocks chronic hypoxia-induced LCMT1 deficiency and tau hyperphosphorylation in rats’ hippocampus. (**a**,**b**) Hippocampal immunohistochemical staining of pS396 and quantitative analysis in control (Group 1), hypoxia (Group 2), siHIF-1α+hypoxia (Group 3), and siHIF-1α+normoxia (Group 4). (**c**) Western blots for tau phosphorylation levels at pS199, pS262, pS396, pS404 sites, and total tau (tau5) in the rat hippocampus (**d**,**e**). Quantification of the relative protein expression levels of pS199, pS262, pS396, pS404, and tau5 after normalization to the β-actin signal. (**f**–**j**) Western blots and quantitative analysis of the T-PP2Ac, M-PP2Ac, LCMT1, and PME-1 in the rat hippocampus. (**k**) PP2A activity assay in different groups of the rat hippocampus. Data represent mean ±SD, *n* = 3 rats per group, * *p* < 0.05, ** *p* < 0.01, *** *p* < 0.001 vs. Group 1.

**Figure 7 ijms-23-16140-f007:**
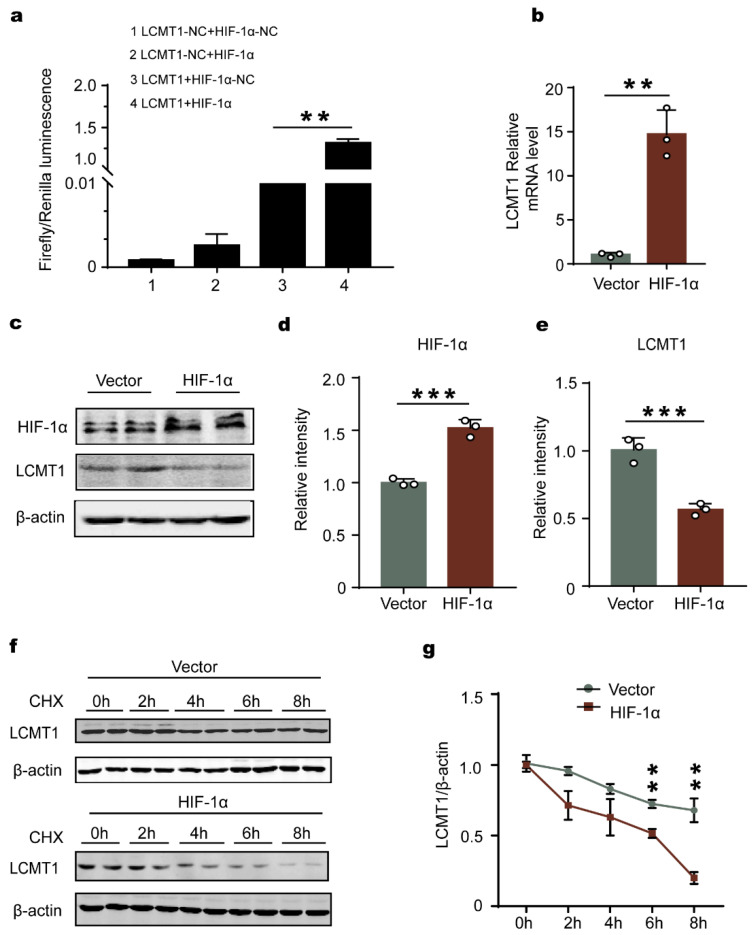
HIF-1α promotes the LCMT1 mRNA expression and accelerates its degradation in C6 cells. (**a**) Dual-luciferase assay. (NC: negative control). (**b**) LCMT1 mRNA expression levels. (**c**–**e**) Vector or HIF-1α plasmid was transfected to C6 cells, Western blots, and quantitative analysis of HIF-1α and LCMT1. (**f**,**g**) Vector or HIF-1α plasmid was transfected into C6 cells treated with translation inhibitor CHX (100 µg/mL) for 2, 4, 6, and 8 h. The degradation of LCMT1 was evaluated by Western blotting and quantitative analysis. Data are represented as mean ±SD, n = 3, ** *p* < 0.01 and *** *p* < 0.001 vs. vector.

**Figure 8 ijms-23-16140-f008:**
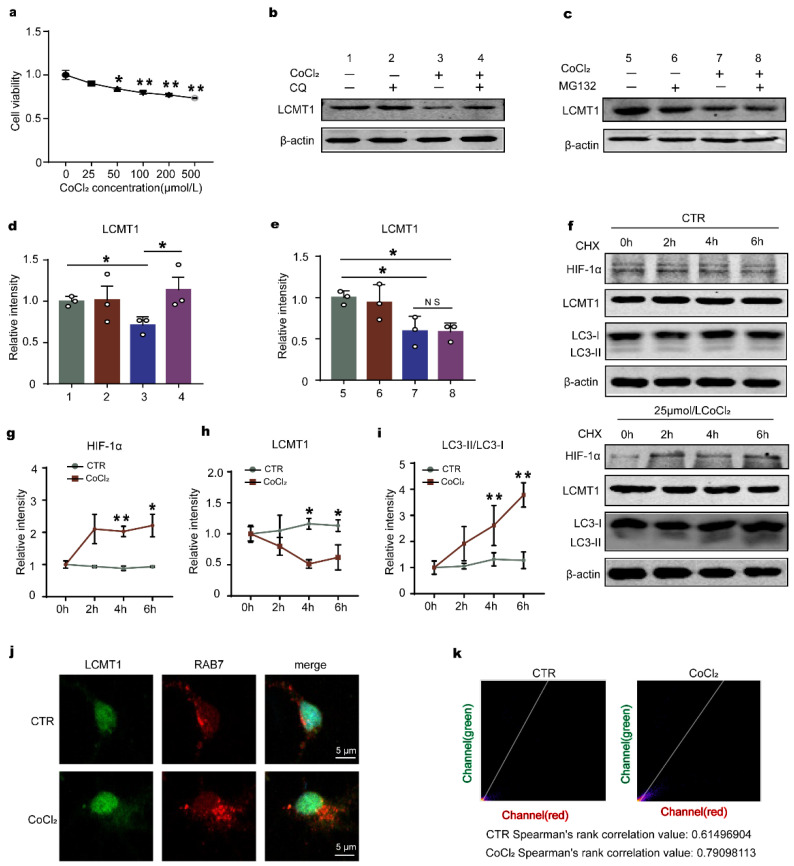
CoCl_2_-induced chemical hypoxia causes degradation of LCMT1 protein through the autophagy lysosomal pathway. (**a**) CCK8 assay in primary hippocampal neurons treated with increasing doses of CoCl_2_ (0, 25, 50, 100, 200, 500 µmol/L) for 8 h. (**b**,**c**) Western blot for LCMT1 expression from neurons treated with the CQ (10 μmol/L) or MG132 (5 μmol/L) with or without simultaneous 25 μmol/L CoCl_2_ treatment. (**d**,**e**) Quantitative analysis of LCMT1 after being treated with CQ or MG132. (**f**) Western blot for HIF-1α, LCMT1, and LC3 in the neurons treated with translation inhibitor CHX (100 µg/mL) for another 0, 2, 4, and 6 h. (**g**–**i**) The protein levels of HIF-1α, LCMT1, and LC3 were quantified. (**j**) LCMT1 and RAB7 expression were detected by immunofluorescence in cells treated with 25 μmol/L CoCl_2_. (**k**) Image J quantification of the average fluorescence values of single cells was used. Data represent mean ±SD, n = 3, * *p* < 0.05, ** *p* < 0.01 vs. control.

**Table 1 ijms-23-16140-t001:** Primary antibodies.

Antibodies	Specific	Type	Dilution	Source
HIF-1α	Anti-HIF-1 alpha	pAb	1:200 for WB	Abcam (ab179483) (Shanghai, China)
pS199	Anti-Phosphorylated tau at Ser199	pAb	1:1000 for WB	Thermo Fisher (44–734G) (Waltham, MA, USA)
pS262	Anti-Phosphorylated tau at Ser262	pAb	1:1000 for WB	Signalway Antibody (#21100) (Jiangsu, China)
pS396	Anti-Phosphorylated tau at Ser396	pAb	1:1000 for WB 1:200 for IHC	Signalway Antibody (#21093)
pS404	Anti-Phosphorylated tau at Ser404	pAb	1:1000 for WB	Signalway Antibody (#21001)
Tau-5	Anti-Total tau	mAb	1:1000 for WB	Millipore (577801) (Burlingtun, MA, USA)
M-PP2Ac	Anti-Methylated PP2Ac at Leu309	mAb	1:500 for WB	Millipore (041479)
T-PP2Ac	Anti-PP2A catalytic subunit	pAb	1:1000 for WB	Cell Signaling (#2038) (Danvers, MA, USA)
S9-GSK3β	Anti-Phosphorylated GSK-β at Ser 9	pAb	1:1000 for WB	Upstate (AP0039) (Thermofisher)
GSK3β	Anti-Glycogen synthase kinase-3β	pAb	1:1000 for WB	Signalway Antibody (40989)
β-actin	Anti-β-actin	pAb	1:1000 for WB	Abcam (ab8227)
LCMT1	Anti-Leucine carboxyl methyltransferase1	pAb	1:1000 for WB	Cell Signaling (#5691)
PME-1	Anti-Protein phosphatase methylesterase1	mAb	1:1000 for WB	Cell Signaling (#29135)
LC3B	Anti-LC3B	pAb	1:1000 for WB 1:200 for IF	Abcam(ab192890)
Rab7	Anti- Rab7	mAb	1:200 for IF	Cell Signaling (#9367)

mAb, monoclonal antibody; pAb, polyclonal antibody; WB, Western blotting; IHC, Immunohistochemistry; IF, Immunofluorescence

**Table 2 ijms-23-16140-t002:** List of primers for RT-qPCR.

Gene Name	NCBI No.	Primer	
HIF-1α	024359.2	(5–3) (3–5)	ACACACAGAAATGGCCCAGTG AATCAGCACCAAGCACGTCA
LCMT1	199405.3	(5–3) (3–5)	GTTGAATGGGTGGGAGACGG TTATCTCCTTCAAACCCAGCTC
GAPDH	017008.4	(5–3) (3–5)	GAAGGTCGGTGTGAACGGAT CCCATTTGATGTTAGCGGGAT

## Data Availability

All data used in this study are available from the corresponding authors on reasonable request.

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
