# Peer review of "HIF-1α Causes LCMT1/PP2A Deficiency and Mediates Tau Hyperphosphorylation and Cognitive Dysfunction during Chronic Hypoxia"

_ijms, 2022, doi:10.3390/ijms232416140_

Round 1
Reviewer 1 Report
In this work Ling Lei et al. the authors evaluated the role of HIF-1α in tau pathology. They found that global hypoxia induced cognitive impairments, HIF-1α increase, tau hyperphosphorylation, and protein phosphatase 2A (PP2A) deficiency with LCMT1 decrease, which were blocked by HIF-1α silencing. They also demonstrate that HIF-1α obviously accelerates LCMT1 degradation counteracting its transcriptional expression, which downregulates PP2A activity, finally resulting in tau hyperphosphorylation and cognitive dysfunction. Overall, this study was well carried out and results fully support the conclusion. Since this study provides a novel molecular mechanism for the chronic hypoxia-related tau pathology, I would highly support the publication of this study by International Journal of Molecular Sciences.
Minor points:
1. Hypoxia is the most common pathological process resulting from inadequate oxygen supply to tissue or the inability to utilize oxygen by mitochondria. In the current study, the author mainly investigate the effect of chronic hypoxia in tau pathology and AD, please also briefly discuss the role of acute hypoxia in AD.
2. The authors should explain why they employ CoCl2 to induce chemical hypoxia.
3. In part of “2.1. Chronic hypoxia impairs cognitive functions in rats”, what is the oxygen concentration of normoxia? Is it same with that in 2.5. as 21% oxygen concentration?
4. In Figure 7g, “Relative intensity” should be changed to LCMT1/β-actin”.
Author Response
Dear Reviewer 1#:
We are very grateful to you for kindly giving us the opportunity to revise our manuscript. We also greatly appreciated your constructive critiques and helpful suggestions. All revisions are highlighted in yellow color. The summary of our revisions and the point-by-point answers to the criticisms are as follows:
- Hypoxia is the most common pathological process resulting from inadequate oxygen supply to tissue or the inability to utilize oxygen by mitochondria. In the current study, the author mainly investigate the effect of chronic hypoxia in tau pathology and AD, please also briefly discuss the role of acute hypoxia in AD.
Answer: We greatly appreciate the suggestion! We have discussed the role of acute hypoxia in AD in the first paragraph of the discussion part. Please see lines 386-390.
- The authors should explain why they employ CoCl2 to induce chemical hypoxia.
Answer: Following this suggestion, we have described the reason CoCl2 was employed in the current study in the revised manuscript. Please see lines 188-191.
CoCl2, which promotes the stabilization of HIF-1α, is commonly used as a drug for chemical hypoxia model of neurodegeneration. Global hypoxia in animals can affect the expression of a variety of substances, and the chemical hypoxia induced by CoCl2 specifically changes HIF-1α”
- In part of “2.1. Chronic hypoxia impairs cognitive functions in rats”, what is the oxygen concentration of normoxia? Is it same with that in 2.5. as 21% oxygen concentration?
Answer: In part of “2.1. Chronic hypoxia impairs cognitive functions in rats”, the oxygen concentration of normoxia is 21%. It is same with that in 2.5. We have rewritten it as being appeared at the first time. Please see line 92.
- In Figure 7g, “Relative intensity” should be changed to LCMT1/β-actin”.
Answer: We really appreciate the suggestion. We have changed “Relative intensity” to “LCMT1/β-actin” in Figure 7g. Please see line 336.

Reviewer 2 Report
The manuscript by Lei et al., investigated changes to HIF1a activity during chronic daily hypoxia exposure. There are interesting concepts and a link to LCMT2 and PP2A activity, and the potential for it to result in changes to tau phosphorylation, but also some experimental and analysis weaknesses, meaning that the conclusion that “rats, chronic hypoxia induces cognitive impairments, and an increase in HIF-1α, tau hyperphosphorylation, and Protein Phosphatase 2A (PP2A) deficiency with a decrease of leucine carboxyl methyltransferase 1 (LCMT1, responsible for PP2A methylation and activity) in the hippocampus” is not supported.
1. The major weakness is the quality of the western blotting whereby the blots presented often do not represent the graphs that are quantified, and with the small experimental sizes (n=3) the error bars are very small for the observed variation. Furthermore, its not clear which bands and how uneven bands have been quantified accurately given the level of background. It was appreciated to have full gels including replicas in the supplementary information. Why are some blots upside down?
The standard (actin) is from a different gel Best practice is for LICOR or similar with a longer dynamic range, and to probe the membrane for a control ( e.g. phosphorylation, or loading control on the same blot with a different secondary antibody). Was this done?
Without this, the quantifications are not considered robust – especially if there is no obvious visual difference in the images. Therefore much of the manuscript is difficult to feel certain about regarding the conclusions.
introduction
2. less than 1% of AD is familial – there is no citation for this figure which I believe is incorrect.
3. Citing that Khalid Iqbal as the first and only person to propose AD is due to tau and Abeta is also not accurate.
4. Hypoxia is not considered a high-risk factor for AD – it has a low risk but is highly prevalent. Please reference recent human studies.
5. Line 60 pg 2 it is not clear what a ‘Ab clearance disorder is’ – is this a disease, or just a change in the rates of clearance due to some change in NEP – which is also not the only explanation for changes in Abeta levels.
6. Other relevant literature is not cited in the introduction e.g. intermittent hypoxia and Abeta levels in AD mice from 2017.
Results
7. Fig 1 shows convincingly that the rats exposed to hypoxia have worse performance in finding the trained platform. Fig 1e. was your MWM actually square? Were the mice released from different positions each trial or day?
8. Fig 2 is unconvincing. The quality of the Tau staining in Fig 2a is poor and does not represent classic NFT staining which is typically in CA1 pyramidal neurons. The number of replicate experiments is not stated and the change in band density is problematic as described above. The changes may be significant but its unclear that they are biologically meaningful. What is the PP2A activity test actually measuring?
9. Sup Fig 1 should be primary data. A robust change is expected?.
10. Similarly the primary neuron experiments ( sup Fig 2) with endogenous tau expression levels would be expected to be more meaningful than the glioma cell line (Fig 3) where tau is over expressed yet are only supplementary. Sup fig 2d is the most convincing of all the blots presented! Also Fig 2j
11. While the levels of HIF1a should increase with hypoxia, the nuclear translocation may be a better readout of function. Could you do sub-cellular fractionation of cells in culture to improve sensitivity? Or immunocytochemistry of the cultured cells?
12. In the in vitro work Fig 3 /4the western blotting is slightly more convincing. But the changes do not appear to be robust with controls being run on different gels. Furthermore, epitopes in some experiments bind 3 bands and only 1 in others. I'm afraid I have to disagree with the conclusion that the data demonstrate that HIF1a expression is positively associated with tau hyperphosphorylation
13. siHIF1a shows less than 50% downregulation. This suggests that the siRNA is not good at its task – 3 siRNAs were trialled but a control siRNA is also important. A different treatment could be to use 2ME2 a HIF1a inhibitor that prevents nuclear trafficking and preventing and change in transcription. Again the variability of the WB with N=3 (two shown) is not believable. N for blots is not shown.
14. Fig 5 5 microlitres is a large volume per ventricle. – confirm if this was unilateral or bi-lateral. And that a control vector was injected. How much of the hippocampus/ cortex/brain was infected? Which tissue was used for the Fig 5b blots? The novel object recognition and other behavioral outcomes are surprisingly robust.
15. Fig 6 see the same issues as above. The error bars are not representative of the blots shown e.g group 1. has more internal variation in shown band density for LCM2 n=3 than the change seen in group 2 n=2 which is not reflected in the quantification.
16. Fig 7 concludes that both over and under-expression of HIF1 upregulates LCMT1 transcription but causes protein degradation. What is '-NC' in Fig 7a?
17. Fig 8 f-I quantification does not match the blots.
Author Response
Dear Reviewer 2#:
We are very grateful to you for kindly giving us the opportunity to revise our manuscript. We also greatly appreciated your constructive critiques and helpful suggestions. All revisions are highlighted in yellow color. The summary of our revisions and the point-by-point answers to the criticisms are as follows:
Comments and Suggestions for Authors
The manuscript by Lei et al., investigated changes to HIF1a activity during chronic daily hypoxia exposure. There are interesting concepts and a link to LCMT2 and PP2A activity, and the potential for it to result in changes to tau phosphorylation, but also some experimental and analysis weaknesses, meaning that the conclusion that “rats, chronic hypoxia induces cognitive impairments, and an increase in HIF-1α, tau hyperphosphorylation, and Protein Phosphatase 2A (PP2A) deficiency with a decrease of leucine carboxyl methyltransferase 1 (LCMT1, responsible for PP2A methylation and activity) in the hippocampus” is not supported.
1.The major weakness is the quality of the western blotting whereby the blots presented often do not represent the graphs that are quantified, and with the small experimental sizes (n=3) the error bars are very small for the observed variation.
Answer: Thank you very much for the expertise of the reviewer and the suggestions. We have re-quantified all the blot bands and changed the standard error into standard deviation. Pleases see the revised manuscript!
Furthermore, its not clear which bands and how uneven bands have been quantified accurately given the level of background.
Answer: Each band was individually framed and quantified for analysis. We have described it in Western blotting of Materials and Methods Section. Please see lines 600-601.
It was appreciated to have full gels including replicas in the supplementary information. Why are some blots upside down?
Answer: Our apologies for this omission. We have corrected them. Please see the revised supplementary information.
The standard (actin) is from a different gel Best practice is for LICOR or similar with a longer dynamic range, and to probe the membrane for a control (e.g. phosphorylation, or loading control on the same blot with a different secondary antibody). Was this done?
Without this, the quantifications are not considered robust – especially if there is no obvious visual difference in the images. Therefore much of the manuscript is difficult to feel certain about regarding the conclusions.
Answer: Thank you very much for the expertise of the reviewer and the suggestions. We didn't actually do this experiment. Following this suggestion, we carried out it with some of antibodies, such as pS199, pS396, HIF-1α or loading control (β-actin). And also we performed it when we used 2ME2 a HIF1a inhibitor as additional experiments added for revision (Figure 4j). Please see the revised supplementary information.
2.less than 1% of AD is familial – there is no citation for this figure which I believe is incorrect.
Answer: Our apologies! We have cited reference and changed it to “less than 5% of AD is familial”. Please see lines 41-43.
3.Citing that Khalid Iqbal as the first and only person to propose AD is due to tau and Abeta is also not accurate.
Answer: Following this suggestion of the reviewer, we have revised it and added the reference (1-3). Please see the first paragraph of the revised Introduction Section.
Reference:
- Iqbal K, Grundke-Iqbal I. Metabolic/signal transduction hypothesis of Alzheimer's disease and other tauopathies. Acta Neuropathol. 2005;109(1):25-31.
- Busche MA, Hyman BT. Synergy between amyloid-beta and tau in Alzheimer's disease. Nat Neurosci. 2020;23(10):1183-93.
- Congdon EE, Sigurdsson EM. Tau-targeting therapies for Alzheimer disease. Nat Rev Neurol. 2018;14(7):399-415.
4.Hypoxia is not considered a high-risk factor for AD – it has a low risk but is highly prevalent. Please reference recent human studies.
Answer: Thank you very much for the expertise of the reviewer and the suggestions. Following this suggestion of the reviewer, we have revised it and added reference recent human studies (4-6).
Reference:
- Leng Y, McEvoy CT, Allen IE, Yaffe K. Association of Sleep-Disordered Breathing With Cognitive Function and Risk of Cognitive Impairment: A Systematic Review and Meta-analysis. JAMA Neurol. 2017 Oct 1;74(10):1237-1245. doi: 10.1001/jamaneurol.2017.2180. Erratum in: JAMA Neurol. 2018 Jan 1;75(1):133. PMID: 28846764; PMCID: PMC5710301.
- Kazim SF, Sharma A, Saroja SR, Seo JH, Larson CS, Ramakrishnan A, Wang M, Blitzer RD, Shen L, Peña CJ, Crary JF, Shimoda LA, Zhang B, Nestler EJ, Pereira AC. Chronic Intermittent Hypoxia Enhances Pathological Tau Seeding, Propagation, and Accumulation and Exacerbates Alzheimer-like Memory and Synaptic Plasticity Deficits and Molecular Signatures. Biol Psychiatry. 2022 Feb 15;91(4):346-358. doi: 10.1016/j.biopsych.2021.02.973. Epub 2021 Mar 24. PMID: 34130857; PMCID: PMC8895475.
- Yaffe K, Laffan AM, Harrison SL, Redline S, Spira AP, Ensrud KE, Ancoli-Israel S, Stone KL. Sleep-disordered breathing, hypoxia, and risk of mild cognitive impairment and dementia in older women. JAMA. 2011 Aug 10;306(6):613-9. doi: 10.1001/jama.2011.1115. PMID: 21828324; PMCID: PMC3600944.
5.Line 60 pg 2 it is not clear what a ‘Ab clearance disorder is’ – is this a disease, or just a change in the rates of clearance due to some change in NEP – which is also not the only explanation for changes in Abeta levels.
Answer: Our apologies for this omission. We have corrected it to “hypoxia leads to an impairment of Aβ clearance via inhibition of NEP”. It is true that some change in NEP is not the only explanation for changes in Abeta levels. Thank you very much for this correction! Please see lines 59-60.
- Other relevant literature is not cited in the introduction e.g. intermittent hypoxia and Abeta levels in AD mice from 2017. Chronic intermittent hypoxia/reoxygenation facilitate amyloid-β generation in mice
Answer: We really appreciate the suggestion. We have added other relevant literature (7) in the introduction of the revised manuscript!
Reference:
- Ryou MG, Mallet RT, Metzger DB, Jung ME. Intermittent hypoxia training blunts cerebrocortical presenilin 1 overexpression and amyloid-β accumulation in ethanol-withdrawn rats. Am J Physiol Regul Integr Comp Physiol. 2017 Jul 1;313(1):R10-R18. doi: 10.1152/ajpregu.00050.2017. Epub 2017 May 10. PMID: 28490448; PMCID: PMC5538853.
Results
7.Fig 1 shows convincingly that the rats exposed to hypoxia have worse performance in finding the trained platform. Fig 1e. was your MWM actually square? Were the mice released from different positions each trial or day?
Answer: Our apologies for this omission. Our MWM is circular and the rat released from same positions each trial or day. We have changed them. Please see the revised Figure 1e. Thank you very much for this observation!
8.Fig 2 is unconvincing. The quality of the Tau staining in Fig 2a is poor and does not represent classic NFT staining which is typically in CA1 pyramidal neurons. The number of replicate experiments is not stated and the change in band density is problematic as described above. The changes may be significant but its unclear that they are biologically meaningful. What is the PP2A activity test actually measuring?
Answer: Thank you very much for this opinion. It is right classic NFT staining is typically in CA1 pyramidal neurons. Some studies (ref 8-10) also showed tau hyperphosphorylation in CA3 neurons in AD. In current study, we found tau hyperphosphorylation in CA3 of the hippocampus. Actually, NFT may be difficult to induce and form in sporadic animal model.
The number of samples each group is three. We have described it in the revised figure legend.
Due to tau hyperphosphorylation linked to neuronal insult and cognitive impairments (ref 11, 12), our result implied that chronic hypoxia-caused tau hyperphosphorylation is associated with cognitive impairments.
PP2A activity (Figure 2h) was evaluated by the release of phosphate per μg protein and per minute (p mol/μg/min), which could be found in “materials and methods”. Please see lines 613-627.
Reference:
- Chai GS, Feng Q, Wang ZH, Hu Y, Sun DS, Li XG, Ke D, Li HL, Liu GP, Wang JZ. Downregulating ANP32A rescues synapse and memory loss via chromatin remodeling in Alzheimer model. Mol Neurodegener. 2017 May 4;12(1):34. doi: 10.1186/s13024-017-0178-8.
- Liu SJ, Zhang JY, Li HL, Fang ZY, Wang Q, Deng HM, Gong CX, Grundke-Iqbal I, Iqbal K, Wang JZ. Tau becomes a more favorable substrate for GSK-3 when it is prephosphorylated by PKA in rat brain. J Biol Chem. 2004 Nov 26;279(48):50078-88. doi: 10.1074/jbc.M406109200. Epub 2004 Sep 15. PMID: 15375165.
- Pei JJ, Sersen E, Iqbal K, Grundke-Iqbal I. Expression of protein phosphatases (PP-1, PP-2A, PP-2B and PTP-1B) and protein kinases (MAP kinase and P34cdc2) in the hippocampus of patients with Alzheimer disease and normal aged individuals. Brain Res. 1994 Aug 29;655(1-2):70-6. doi: 10.1016/0006-8993(94)91598-9.
- Kandimalla R, Manczak M, Yin X, Wang R, Reddy PH. Hippocampal phosphorylated tau induced cognitive decline, dendritic spine loss and mitochondrial abnormalities in a mouse model of Alzheimer's disease. Hum Mol Genet. 2018;27(1):30-40.
- Raz L, Bhaskar K, Weaver J, Marini S, Zhang Q, Thompson JF, et al. Hypoxia promotes tau hyperphosphorylation with associated neuropathology in vascular dysfunction. Neurobiol Dis. 2019; 126:124-36.
9.Sup Fig 1 should be primary data. A robust change is expected?.
Answer: Following this suggestion, we have put original Sup Fig 1 into figure 3a-c.
In the presence of oxygen, HIF-1 is degraded by ubiquitin proteases. With hypoxia, the HIF-1α subunit cannot be degraded and accumulates, and binds to HIF-1β to form dimers, which are then transferred to the nucleus and activate genes encoding proteins involved in hypoxic homeostatic response. Therefore, HIF-1α level is expectedly increased in the chronic hypoxia.
10.Similarly the primary neuron experiments (sup Fig2) with endogenous tau expression levels would be expected to be more meaningful than the glioma cell line (Fig 3) where tau is over expressed yet are only supplementary. Sup fig 2d is the most convincing of all the blots presented! Also Fig 2j.
Answer: Thank you very much for the expertise of the reviewer and the suggestions. Following this suggestion of the reviewer, we have put original supplementary Figure 2 into revised Figure 3d-o, and placed original Figure 3 to revised supplementary Figure 2. Please see the revised manuscript.
11.While the levels of HIF1a should increase with hypoxia, the nuclear translocation may be a better readout of function. Could you do sub-cellular fractionation of cells in culture to improve sensitivity? Or immunocytochemistry of the cultured cells?
Answer: Following this suggestion of the reviewer, we have carried out immunofluorescence of primary hippocampus neurons and found that hypoxia induced a marked nuclear translocation of HIF-1α, further supporting that the levels of HIF-1a increases with hypoxia. Please see the supplemental Figure 1.
12.In the in vitro work Fig 3/4 the western blotting is slightly more convincing. But the changes do not appear to be robust with controls being run on different gels. Furthermore, epitopes in some experiments bind 3 bands and only 1 in others. I'm afraid I have to disagree with the conclusion that the data demonstrate that HIF1a expression is positively associated with tau hyperphosphorylation
Answer: Thank you very much for your observation. Actually, the original figure 3 western blotting were resulted from the rat C6 glioma cells stably expressing human full-length tau441, while the figure 4 western blotting were resulted from the rat endogenous neuronal tau. Human derived tau exhibits only 1 band and rat tau exhibits 3 bands (ref 13, 14).
Reference:
- Mendoza J, Sekiya M, Taniguchi T, Iijima KM, Wang R, Ando K. Global analysis of phosphorylation of tau by the checkpoint kinases Chk1 and Chk2 in vitro. J Proteome Res. 2013 Jun 7;12(6):2654-65. doi: 10.1021/pr400008f. Epub 2013 Apr 26. PMID: 23550703; PMCID: PMC3757556.
- Lionnet A, Wade MA, Corbillé AG, Prigent A, Paillusson S, Tasselli M, Gonzales J, Durieu E, Rolli-Derkinderen M, Coron E, Duchalais E, Neunlist M, Perkinton MS, Hanger DP, Noble W, Derkinderen P. Characterisation of tau in the human and rodent enteric nervous system under physiological conditions and in tauopathy. Acta Neuropathol Commun. 2018 Jul 23;6(1):65. doi: 10.1186/s40478-018-0568-3. PMID: 30037345; PMCID: PMC6055332.
Reference13 Mendoza J et al. J Proteome Res. 2013
13.siHIF1a shows less than 50% downregulation. This suggests that the siRNA is not good at its task – 3 siRNAs were trialled but a control siRNA is also important. A different treatment could be to use 2ME2 a HIF1a inhibitor that prevents nuclear trafficking and preventing and change in transcription. Again the variability of the WB with N=3 (two shown) is not believable. N for blots is not shown.
Answer: We greatly appreciate these concerns. Actually, we firstly constructed three siHIF-1α plasmids, and chose the plasmid (6315) with the highest efficiency for virus packaging. A control siRNA is important. Following the suggestion of the review, we employed 2ME2 a HIF1α inhibitor to downregulate HIF1α and detected p-tau, PP2Ac, LCMT1 and HIF1α, which is consistent with the results from siHIF1a. Please see the revised Figure 4j-n.
The number of samples each group is three. We have described it in the revised figure legend. In the future work, we will use more samples. Thank you very much for your suggestion!
14.Fig 5 5 microlitres is a large volume per ventricle. – confirm if this was unilateral or bi-lateral. And that a control vector was injected. How much of the hippocampus/ cortex/brain was infected? Which tissue was used for the Fig 5b blots? The novel object recognition and other behavioral outcomes are surprisingly robust.
Answer: Thank you very much for this concern. In our experiment, 5 microlitres was bi-lateral. We have described it in the Materials and Methods Sections at line 501. Please see the revised manuscript. Almost 100% the hippocampus was infected. Hippocampus was used for the Fig 5b blots. In the current study, we really observed the obvious outcomes from behavioral tests.
- Fig 6 see the same issues as above. The error bars are not representative of the blots shown e.g group 1. has more internal variation in shown band density for LCM2 n=3 than the change seen in group 2 n=2 which is not reflected in the quantification.
Answer: Thank you very much for the expertise of the reviewer and the suggestions. We have re-quantified all the blot bands and changed the standard error into standard deviation. Pleases see the revised manuscript!
The number of samples each group is three. We have described it in the revised figure legend. In the future work, we will use more samples. Thank you very much for your suggestion!
16.Fig 7 concludes that both over and under-expression of HIF1 upregulates LCMT1 transcription but causes protein degradation. What is '-NC' in Fig 7a?
Answer: Thank you very much for your observation. '-NC' in Fig 7a is the abbreviation for “negative control”. We have described it in the figure legend. Please see the revised manuscript at line 340.
17.Fig 8 f-I quantification does not match the blots.
Answer: We re-quantified Figure 8f. In figure 8g-i (HIF-1α, LCMT1, LC3-II/LC-3I), all the relative intensity for 0 hour were set to 1. Please see the revised manuscript!

Round 2
Reviewer 2 Report
• The quality of the presentation with individual dots and SD for western blots has improved the transparency of the data.
• The addition of the 2ME2-treatment data provides additional support for the proposed mechanistic pathway. However, as it is an inhibitor of nuclear translation of HIF1a I was surprised the expression levels dropped.
•Although I remain unconvinced regarding some of the conclusions, other readers can now make their own judgments.
further minor clarifications: please clarify whether the n=3 is one experiment set up in triplicate, 3 experiments ( set up on different days with different cells/passages/dissections).
please state 2.5ul per injection site not 5ul in total.
please provide evidence of the expression of the virus in the hippocampus given the statement it was all - did that also include cortex or hypothalamus?
I am impressed that the experimental revision was completed in 15 days!
Author Response
Dear Reviewer:
We are very grateful to you for kindly giving us the opportunity to revise our manuscript. We also greatly appreciated your constructive critiques and helpful suggestions. All revisions are highlighted in yellow color. The summary of our revisions and the point-by-point answers to the criticisms are as follows:
1.The quality of the presentation with individual dots and SD for western blots has improved the transparency of the data.
Answer: Thank you so much for your constructive evaluation about the changes of the presentation for western blots. We really also appreciate your suggestion for making these changes.
2.The addition of the 2ME2-treatment data provides additional support for the proposed mechanistic pathway. However, as it is an inhibitor of nuclear translation of HIF1α I was surprised the expression levels dropped.
Answer: Thank you very much for this query.
In normoxia, hydroxylation of two proline residues and acetylation of a lysine residue at the oxygen-dependent degradation domain of HIF-1α trigger its association with pVHL E3 ligase complex, leading to HIF-1α degradation via ubiquitin-proteasome pathway. In hypoxia, the HIF-1α subunit cannot be degraded and accumulates, and binds to HIF-1β to form dimers, which are then transferred to the nucleus and activate genes encoding proteins involved in hypoxic homeostatic response.
In the current study, we used 2ME2, an inhibitor of nuclear translation of HIF1a and found that 2ME2-treatment induces a decrease in HIF1a level. We suppose that 2ME2-treatment blocks the accumulation of the HIF-1α in nucleus, leading to most of the HIF-1α to be located in cytoplasm and therefore easily degraded. Some previous studies also showed similar results (ref 1-4). However, the exact mechanism of HIF-1α downregulation by 2ME2 remains to be further studies.
Reference:
[1] Mabjeesh NJ, Escuin D, LaVallee TM, Pribluda VS, Swartz GM, Johnson MS, Willard MT, Zhong H, Simons JW, Giannakakou P. 2ME2 inhibits tumor growth and angiogenesis by disrupting microtubules and dysregulating HIF. Cancer Cell. 2003 Apr;3(4):363-75.
[2] Zheng S, Ni J, Li Y, Lu M, Yao Y, Guo H, Jiao M, Jin T, Zhang H, Yuan A, Wang Z, Yang Y, Chen Z, Wu H, Hu W. 2-Methoxyestradiol synergizes with Erlotinib to suppress hepatocellular carcinoma by disrupting the PLAGL2-EGFR-HIF-1/2α signaling loop. Pharmacol Res. 2021 Jul; 169:105685.
[3] Qian L, Rawashdeh O, Kasas L, Milne MR, Garner N, Sankorrakul K, Marks N, Dean MW, Kim PR, Sharma A, Bellingham MC, Coulson EJ. Cholinergic basal forebrain degeneration due to sleep-disordered breathing exacerbates pathology in a mouse model of Alzheimer's disease. Nat Commun. 2022 Nov 2;13(1):6543.
[4] Zhou D, Matchett GA, Jadhav V, Dach N, Zhang JH. The effect of 2-methoxyestradiol, a HIF-1 alpha inhibitor, in global cerebral ischemia in rats. Neurol Res. 2008 Apr;30(3):268-71.
3.Although I remain unconvinced regarding some of the conclusions, other readers can now make their own judgments.
Answer: Thank you very much for your suggestions and opinions. We will further improve our works in the following experiments.
4.further minor clarifications: please clarify whether the n=3 is one experiment set up in triplicate, 3 experiments (set up on different days with different cells/passages/dissections).
Answer: Our apologies for this omission. We performed western blot experiments on 3 rats in each group. The n=3 is not one experiment set up in triplicate, we actually ran two experiments in two consecutive days, one with two samples and one with one sample. In fact, it is the limitation of our manuscript. We will increase the experimental sizes in our future studies following the suggestion of the reviewer. Thanks again for the suggestion.
5.please state 2.5ul per injection site not 5ul in total.
Answer: Thank you very much for the expertise of the reviewer and the suggestions. Following this suggestion of the reviewer, we have revised it. Please see the line 504.
6.please provide evidence of the expression of the virus in the hippocampus given the statement it was all - did that also include cortex or hypothalamus?
Answer: Intracranial delivery of AAV to rats has been shown to result in widespread and long-lasting neuronal expression of transgenes (ref 5). In our previous study (ref 6), to investigate how extensive and how long expression was induced by the AAV infection of rats, we carried out GFP immunohistochemistry at different periods after infection. As expected, GFP expression was observed in the choroid plexus, cerebral cortex, hippocampus, and ventricular area at 3 wk, 9 wk, 5 mo, and 8 mo after AAV injection studied (Fig. 1F).
Reference:
- Lawlor, P. A., Bland, R. J., Das, P., Price, R. W., Holloway, V., Smithson, L., Dicker, B. L., During, M. J., Young, D., and Golde, T. E. (2007) Novel rat Alzheimer’s disease models based on AAV-mediated gene transfer to selectively increase hippocampal Abeta levels. Mol. Neurodegener. 2, 11
- Wang XC, Blanchard J, Kohlbrenner E, Clement N, Linden RM, Radu A, Grundke-Iqbal I, Iqbal K. The carboxy-terminal fragment of inhibitor-2 of protein phosphatase-2A induces Alzheimer disease pathology and cognitive impairment. FASEB J. 2010; 24(11):4420-4432.
Since 2010, we have been using the AAV delivery system to infect animals for long time and have published a series paper as follows.
- Li L, Jiang Y, Wu G, Mahaman YAR, Ke D, Wang Q, Zhang B, Wang JZ, Li HL, Liu R, Wang XC*. Phosphorylation of Truncated Tau Promotes Abnormal Native Tau Pathology and Neurodegeneration. Mol Neurobiol. 2022 Oct;59(10):6183-6199. doi: 10.1007/s12035-022-02972-7. Epub 2022 Jul
- Wang XC, Blanchard J, Grundke-Iqbal I, Wegiel J, Deng HX, Siddique T, Iqbal K. Alzheimer disease and amyotrophic lateral sclerosis: an etiopathogenic connection. Acta Neuropathol. 2014 Feb;127(2):243-256.
- Guo C, Liu Y, Fang M, Li Y, Li W, Mahaman YAR, Zeng K, Xia Y, Ke D, Liu R, Wang JZ, Shen H, Shu X, Wang XC*. ω-3PUFAs improves cognitive impairments through Ser133 phosphorylation of CREB up-regulating BDNF/TrkB signal in schizophrenia. Neurotherapeutics, 2020 Jul;17(3):1271-1286
- Xu J, Guo C, Liu Y, Wu G, Ke D, Wang Q, Mao J, Wang JZ, Liu R, Wang XC*. Nedd4l downregulation of NRG1 in the mPFC induces depression-like behaviour in CSDS mice. Transl Psychiatry. 2020 Jul 23;10(1):249. doi: 10.1038/s41398-020-00935-x.
- Huang F, Wang M, Liu R, Wang JZ, Schadt E, Haroutunian V, Katsel P, Zhang B, Wang XC*. CDT2 Controlled Cell Cycle Reentry Regulates the Pathogenesis of Alzheimer’s Disease. Alzheimers & Dementia. 2019 Feb;15(2):217-231.
- Qin M, Li H, Bao J, Xia Y, Ke D, Wang Q, Liu R, Wang JZ, Zhang B, Shu X, Wang XC*. SET SUMOylation promotes its cytoplasmic retention and induces tau pathology and cognitiveimpairments. Acta Neuropathol Commun. 2019 Feb 15; 7(1):21. doi: 10.1186/s40478-019-0663-0.
- Zhang Q, Xia Y, Wang Y, Shentu Y, Zeng K, Mahaman YAR, Huang F, Wu M, Ke D, Wang Q, Zhang B, Liu R, Wang JZ, Ye K, Wang XC*. CK2 Phosphorylating I2PP2A/SET Mediates Tau Pathology and Cognitive Impairment. Front Mol Neurosci. 2018 Apr 30;11:146. doi: 10.3389/fnmol.2018.00146. eCollection 2018.
- Liu X, Zeng K, Li M, Wang Q, Liu R, Zhang B, Wang JZ, Shu X, Wang XC*. Expression of P301L-hTau in mouse MEC induces hippocampus-dependent memory deficit. Sci Rep. 2017 Jun 20;7(1):3914. doi: 10.1038/s41598-017-04305-4.
- Wang XC, Blanchard J, Tung YC, Grundke-Iqbal I, Iqbal K*. Inhibition of Protein Phosphatase-2A (PP2A) by I1PP2A Leads to Hyperphosphorylation of Tau, Neurodegeneration and Cognitive Impairment in Rats. J Alzheimers Dis. 2015 Jan 1;45(2):423-435. doi: 10.3233/JAD-142403.
- Yu G, Yan TH, Feng Y, Liu XH, Xia YY, Luo HB, Wang JZ, Wang XC*. Ser9 phosphorylation causes cytoplasmic detention of I2PP2A/SET in Alzheimer disease. Neurobiology of Aging. 2013 Jul;34(7):1748-1758
We really greatly appreciated the constructive critiques and helpful suggestions mentioned above.
We hope that you will find this revised manuscript acceptable for publication in IJMS.
Sincerely,
Hong-Lian Li and Xiaochuan Wang
